# NExT-OMNI: Towards Any-to-Any Omni-modal Foundation Models with Discrete Flow Matching

**Run Luo**[1,2]   **Xiaobo Xia**[3,4*]   **Lu Wang**[5]   **Longze Chen**[1,2]   **Renke Shan**[5]   **Jing Luo**[1,2]
**Min Yang**[1,2,6*]   **Tat-Seng Chua**[3]
[1]Shenzhen Institutes of Advanced Technology, Chinese Academy of Sciences
[2]University of Chinese Academy of Sciences  [3]National University of Singapore
[4]University of Science and Technology of China  [5]Rtizz-AI
[6]Shenzhen University of Advanced Technology
{r.luo,min.yang}@siat.ac.cn, xiaoboxia.uni@gmail.com

## Abstract

Next-generation multimodal foundation models capable of any-to-any cross-modal generation and multi-turn interaction will serve as core components of artificial general intelligence systems, playing a pivotal role in human-machine interaction. However, most existing multimodal models remain constrained by autoregressive architectures, whose inherent limitations prevent a balanced integration of understanding and generation capabilities. Although hybrid and decoupling strategies have been explored to address these tasks within unified frameworks separately, their redundant, non-integrated designs limit their applicability to broader scenarios, such as cross-modal retrieval. In this work, we introduce NExT-OMNI, an open-source omnimodal foundation model that achieves unified modeling through discrete flow paradigms. By leveraging metric-induced probability paths and kinetic optimal velocities, NExT-OMNI natively supports any-to-any understanding and generation with enhanced response efficiency, while enabling broader application scenarios through concise unified representations rather than task-decoupled designs. Trained on large-scale interleaved text, image, video, and audio data, NExT-OMNI delivers competitive performance on multimodal understanding and generation benchmarks, while outperforming prior unified models in multi-turn multimodal interaction and cross-modal retrieval, highlighting its architectural advantages as a next-generation multimodal foundation model. The code is available at https://github.com/ritzz-ai/Next-OMNI.

## 1 Introduction

Unified multimodal understanding and generation has emerged as a critical bottleneck for achieving artificial general intelligence (AGI), attracting growing academic attention (Dong et al., 2024; Ge et al., 2024; Fu et al., 2024; Xie & Wu, 2024). Numerous studies (Wu et al., 2024c; Dong et al., 2024; Ge et al., 2024; Wu et al., 2024b) have attempted to leverage successful autoregressive (AR) techniques from large language models (LLMs) (Touvron et al., 2023; Yang et al., 2024; Wu et al., 2024d) to achieve unified modeling of multimodal understanding and generation. However, these attempts have failed to achieve desired outcomes due to inherent conflicts within AR paradigms when handling understanding and generation tasks (Wu et al., 2024b;c; Dong et al., 2024).

Subsequent works have adopted hybrid architectures (Xie et al., 2024; Zhou et al., 2024) and modular decoupling techniques (Wu et al., 2024a; Deng et al., 2025; Chen et al., 2025a) to separately process understanding and generation tasks within relatively unified frameworks. These methods improve performance on both task categories and narrow the gap between open-source models and closed-source systems (OpenAI, 2025; DeepMind, 2025). Nevertheless, such AR-based hybrid architectures primarily rely on decoupling rather than fusion design. Although effective in under-

---
*Corresponding authors.

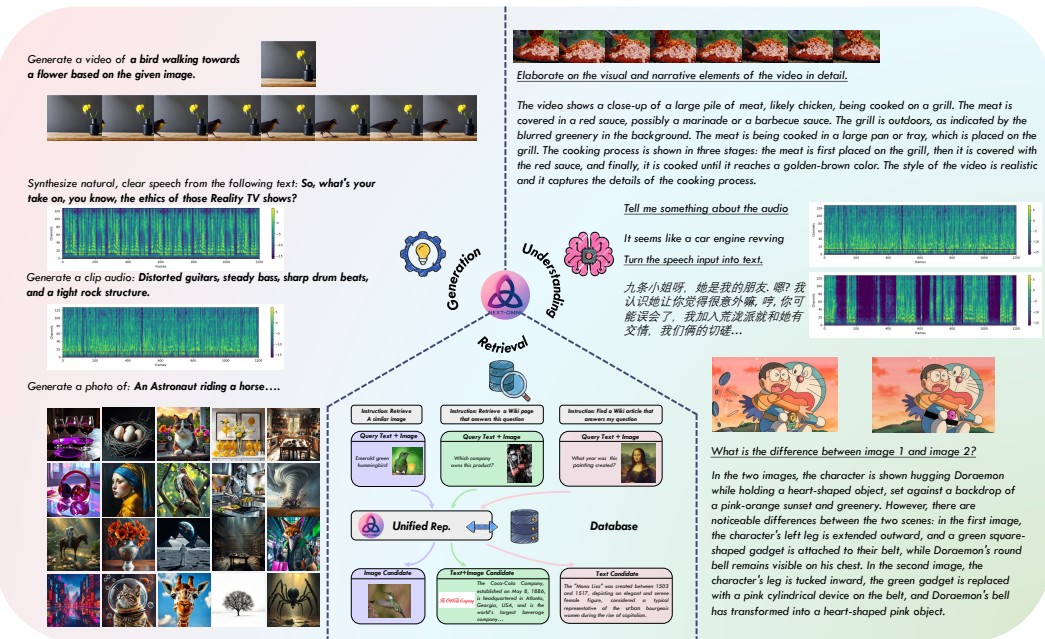

Figure 1: **Overview of the NExT-OMNI framework.** NExT-OMNI is a unified framework for omnimodal tasks, offering strong understanding, generation, and retrieval capabilities. The framework enables any-to-any operations across modalities for both generation and understanding tasks, while achieving accelerated processing. It also supports cross-modal retrieval by leveraging rich multimodal representations aggregated within its architecture.

standing and generation tasks, they inevitably introduce additional parameterized modules, thereby increasing structural complexity and slowing inference. Moreover, this separated design struggles to support tasks that demand deep multimodal feature fusion and flexible any-to-any cross-modal tasks, such as cross-modal retrieval, which limits their applicability to more general multimodal scenarios.

In recent years, models based on discrete diffusion and flow matching have demonstrated competitive advantages over traditional AR counterparts across multiple domains, including language modeling (Nie et al., 2025; Ye et al., 2025a; Fan et al., 2026), image modeling (Rombach et al., 2022a; Li et al., 2024f), and audio modeling (Du et al., 2024; Li et al., 2025c). Unlike sequential AR models, these models begin with completely corrupted sequences and iteratively denoise entire sequences in parallel, enabling richer bidirectional information integration to enhance task performance. In addition, they achieve flexible and controllable generation through inherent iterative refinement processes, while demonstrating accelerated sampling potential through parallel decoding, providing a more promising fusion perspective for any-to-any multimodal understanding and generation tasks. However, this research direction remains underexplored, with its potential in unified understanding, generation, and retrieval yet to be fully realized.

To address this research gap, we propose NExT-OMNI, a fully open-source omnimodal foundation model based on discrete flow matching (DFM) techniques (Shaul et al., 2024), trained on large-scale, carefully curated interleaved multimodal datasets encompassing images, text, video, and audio. As shown in Figure 1, NExT-OMNI achieves faster any-to-any omnimodal generation through a streamlined unified architecture, while enabling more precise cross-modal retrieval through unified representations with intermediate feature fusion. Overall, NExT-OMNI not only demonstrates competitive performance with reduced latency on standard multimodal understanding and generation benchmarks, but also exhibits superior performance in multi-turn multimodal interaction and cross-modal retrieval. These results highlight that DFM-based unified multimodal understanding and generation modeling architectures provide a powerful fusion perspective for advancing multimodal unification with broader applicability.

The contributions of this paper are summarized as follows: 1) We propose NExT-OMNI, the first open-source omnimodal model built entirely on DFM, which is capable of achieving any-to-any generation across text, images, video, and audio with faster inference. 2) We design a reconstruction-

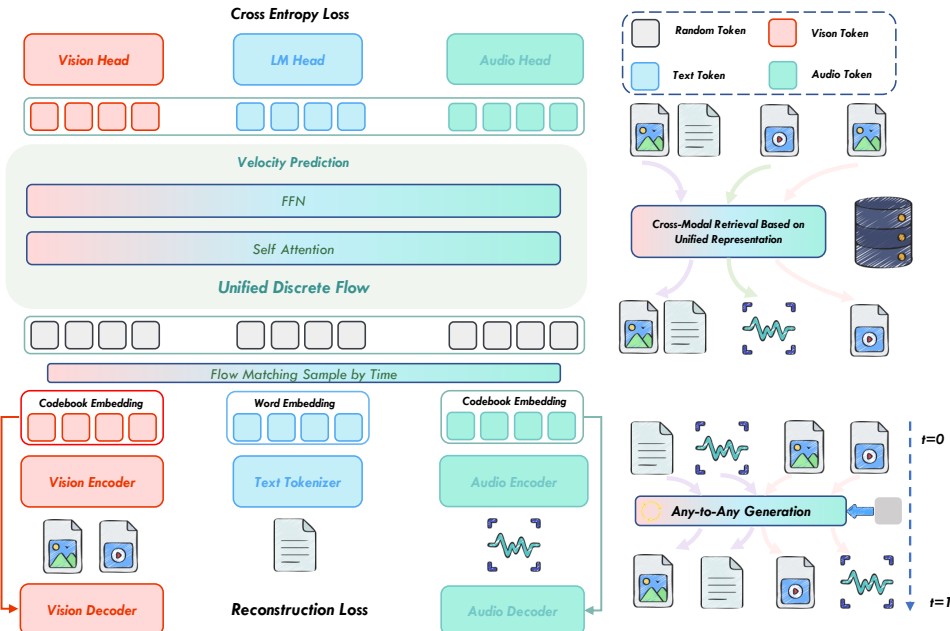

Figure 2: **Pipeline of the NExT-OMNI framework.** NExT-OMNI employs a DFM paradigm for unified omnimodal training, with multimodal self-attention at every layer to deeply fuse information across modalities. Unlike prior methods using multiple encoders or mixture-of-experts, it trains a single encoder simultaneously for understanding and generation, producing unified representations that enable any-to-any multimodal tasks with a streamlined architecture and strong generalization.

enhanced unified representation with intermediate feature fusion. This design not only enables precise cross-modal retrieval but also supports multi-turn any-to-any multimodal interactions, demonstrating advantages over separated AR-based frameworks. 3) We conduct extensive experiments across understanding, generation, and retrieval benchmarks. Results show that NExT-OMNI consistently achieves competitive or superior performance with reduced latency, validating the potential of DFM-based architectures as a promising paradigm for unified multimodal modeling.

## 2 METHOD

### 2.1 ARCHITECTURE

**Modality Encoders**. We design vision and audio encoders grounded in unified representation principles (Luo et al., 2024a; Wu et al., 2024c; Ma et al., 2025a), enabling unified representational modeling. This design allows single-modal encoders to support both generation and understanding tasks, while also mitigating encoder redundancy.

**Backbone**. NExT-OMNI is initialized from the pretrained weights of AR-based LLMs and employs discrete flow matching within a three-stage progressive training framework on carefully curated omnimodal data. Following previous work (Gong et al., 2024b; Ye et al., 2025a; Wang et al., 2025), we retain the shifting operation to output logits by one position during training, enabling our model to inherit the next-token prediction capabilities of AR-based LLMs to the greatest extent possible. To expand model application scenarios such as cross-modal retrieval while streamlining model structure, we utilize deep bidirectional attention feature fusion rather than relying on additional MoE/MoT decoupling mechanisms (Xie et al., 2024; Deng et al., 2025).

**Modality Heads**. Since NExT-OMNI employs discrete flow matching, it eliminates the need for additional diffusion or flow heads (Dong et al., 2024; Luo et al., 2024a; Han et al., 2025; Chen et al., 2025a) specifically designed for generation optimization. Instead, it only requires lightweight heads for discrete token decoding, thereby substantially improving training efficiency and accelerating generation response. Furthermore, we introduce separate modality-specific heads for decoding each type of modality data, rather than extending the language model's vocabulary head directly. This

design effectively preserves text generation capabilities. Additional details on the modality encoders and heads are provided in Appendix D.

## 2.2 Unified Representation Modeling

We first perform warmup training for unified representation modeling of the modality encoders. Two objectives are combined: (i) a reconstruction loss $\mathcal{L}_{\text{rec}}^{\text{M}}$, implemented with an auxiliary VQVAE (Van Den Oord et al., 2017) quantizer and modality-specific decoders that capture low-level details; (ii) a semantic alignment loss $\mathcal{L}_{\text{sem}}^{\text{M}}$ with an auxiliary text encoder or text decoder that emphasize high-level semantic alignments. Here, the superscript M corresponds to the modality. Given a continuous input vector $z^{\text{M}}$, which is obtained from a modality input $X^{\text{M}}$ encoded by the corresponding modality encoder $\mathcal{E}^{\text{M}}$, the goal is to map it to the closest vector in a learnable codebook $\mathcal{C}^{\text{M}}$. The quantization process is formulated as $z^q = \arg\min_{c \in \mathcal{C}^{\text{M}}} \|z^{\text{M}} - c\|_2$, where $\mathcal{C}^{\text{M}} = \{c_1^{\text{M}}, c_2^{\text{M}}, \ldots, c_K^{\text{M}}\}$, and $K$ is the number of codebook entries. The aim is to map the input $z^{\text{M}}$ to the most representative vector $c_k^{\text{M}} \in \mathcal{C}^{\text{M}}$ via a VQ loss $\mathcal{L}_{\text{VQ}}^{\text{M}}$. Then we use the corresponding modality decoder $\mathcal{D}^{\text{M}}$ to restore the modality input $X^{\text{M}}$ conditioned on the representative vector via a modality restoration loss $\mathcal{L}_{\text{R}}^{\text{M}}$. The reconstruction loss can be formulated as $\mathcal{L}_{\text{rec}}^{\text{M}} = \mathcal{L}_{\text{R}}^{\text{M}} + \mathcal{L}_{\text{VQ}}^{\text{M}} + \mathcal{L}_{\text{G}}^{\text{M}}$, where $\mathcal{L}_{\text{G}}^{\text{M}}$ is a discriminator loss. The total warmup training objective for our unified representation-based modality encoders can be expressed:

$$\mathcal{L}_{\text{total}}^{\text{M}} = \mathcal{L}_{\text{rec}}^{\text{M}} + \mathcal{L}_{\text{sem}}^{\text{M}}, \quad \text{M} \in \{\text{A}, \text{V}\}. \tag{1}$$

In more detail, to address the distinct semantic granularity requirements of vision modality V and audio modality A, we adopt token-level caption generation alignment $\mathcal{L}_{\text{sem}}^{\text{A}} = \mathcal{L}_{\text{cap}}^{\text{A}}$, for the audio encoder, following Whisper (Radford et al., 2023), and sentence-level contrastive semantic alignment $\mathcal{L}_{\text{sem}}^{\text{V}} = \mathcal{L}_{\text{constra}}^{\text{V}}$ for the vision encoder, following CLIP-ViT (Dosovitskiy et al., 2021).

## 2.3 Discrete Flow Matching Modeling

As illustrated in Figure 2, given an omnimodal vision-text-audio sequence input sampled from target distributions $q(\cdot)$, NExT-OMNI first utilizes VQVAE-based modality encoders and text tokenizers to convert it into discrete target token sequences $x_1 = (x_1^1, x_1^2, \ldots, x_1^D)$, where each element $x_1^n$ is arranged in the order in which they appear in the original content. At each training step, a time $t \in [0, 1]$ is uniformly sampled, and a noisy sequence $x_t$ is sampled according to the probability path $p_t(\cdot|x_1)$ defined in Appendix A. Then, the model receives a noisy sequence $x_t$ as an input and predicts $x_1$, outputting per-token logits for each position. Note that, unlike previous methods (Wang et al., 2024b; Han et al., 2025) that directly utilize discrete tokens, we extract continuous representative vector $c_{z^q}^{\text{M}}$ with rich semantic and detailed information from the corresponding quantizer codebooks $\mathcal{C}^{\text{M}}$ of modality encoders based on discrete tokens, and achieve dimensional alignment with text embeddings through lightweight projection. This simple yet effective method enables the model to achieve superior performance in subsequent optimization. During training, we only perform correction training on the response portions of instruction data. The discrete flow matching (DFM) modeling is defined as the expected cross-entropy loss between the ground-truth sequence $x_1$ and the model's predicted distribution as follows:

$$\mathcal{L}_{\text{ce}} = \mathbb{E}_{t \sim \mathcal{U}[0,1], \, x_1 \sim q(\cdot), \, x_t \sim p_t(\cdot|x_1)} \left[ -\sum_{i=1}^{D} \log p_{1|t}\left(x_1^i | x_t\right) \right] \tag{2}$$

In addition to the cross-entropy loss mentioned above, to prevent the model from overly favoring semantic information during DFM training while discarding fine-grained information embedded in the unified representations of modality encoders, which would degrade model performance on understanding and generation tasks, we constrain the DFM training by reusing the corresponding reconstruction losses from the modality encoders in unified representation modeling. This maintains rich and detailed information, which not only improves understanding and generation performance but also enhances deep multimodal feature fusion, providing more precise cross-modal retrieval capabilities. The overall training objective can then be rewritten below:

$$\mathcal{L}_{\text{overall}} = \lambda_1 \cdot \mathcal{L}_{\text{ce}} + \lambda_2 \cdot \mathcal{L}_{\text{rec}}^{\text{V}} + \lambda_3 \cdot \mathcal{L}_{\text{rec}}^{\text{A}}, \tag{3}$$

where $\lambda_1$, $\lambda_2$, and $\lambda_3$ are the coefficient that controls the trade-offs between DFM modeling and the modality reconstruction loss. We adopt the GradNorm (Chen et al., 2018) method to dynamically adjust the coefficient, ensuring equal gradient update contributions to the model during training.

## 2.4 More Details of Training and Inference

During joint training, different data modalities require different training modules due to modality differences, and random mixed training leads to load imbalance that wastes substantial computational resources. To improve training efficiency, we conduct training for only one modality within any given training batch, achieving the joint training objective through interleaved training of multiple tasks and gradient accumulation, effectively enhancing computational resource utilization efficiency and achieving a $1.4\times$ improvement in training efficiency.

As illustrated in Figure 3, to further improve NExT-OMNI's performance on understanding tasks, we design a dynamic length generation strategy. Specifically, during training, we insert additional <PAD> tokens to ensure that response sequences participating in training are multiples of the block size. During inference, leveraging the properties that simple tokens can be determined in a single denoising step, we dynamically adjust to appropriate preset generation lengths in block size increments based on <EOS> confidence, then perform multi-step iterative denoising. This strategy dramatically enhances the model's text generation capabilities at minimal cost. Furthermore, we observe phenomena similar to DLLM (Liu et al., 2025c) during the multi-step denoising process in inference, where most features change minimally throughout the multi-step denoising procedure, providing opportunities for inference acceleration using caching mechanisms. We cache and perform minimal updates on the instruction portion throughout the entire inference process, while adaptively updating during the response process based on the cosine similarity between value features and cached features. This vanilla adaptive cache implementation, combined with the parallel decoding advantages of NExT-OMNI's DFM architecture, achieves a $1.2\times$ inference response speed improvement compared to AR architectures. Our cache acceleration and dynamic generation strategies enable superior performance with faster response speed.

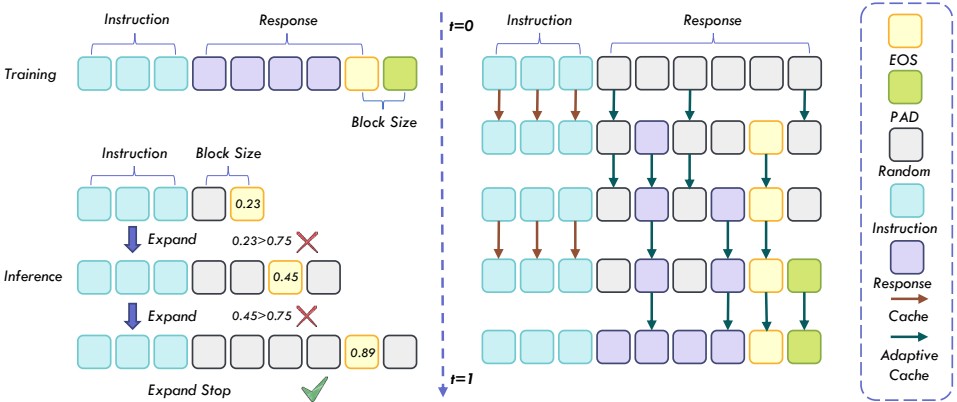

Figure 3: **Illustrations of dynamic generation strategy (left) and vanilla adaptive cache design (right).** During training, responses are padded to multiples of the block size, allowing the model to extend preset response lengths in block-size increments during inference. This improves performance on understanding tasks that require dynamic-length generation. The vanilla adaptive cache caches instruction features and selectively caches response features based on feature cosine similarity, accelerating inference and decoding. Combined with parallel decoding, this simple yet efficient caching design enables NExT-OMNI to generate responses faster than AR-based models.

## 3 Experiments

In this section, extensive experiments are conducted to demonstrate the superiority of the proposed NExT-OMNI and justify our claims. We first introduce implementation details (Section 3.1). Following this, a series of results on various practical tasks and subsequent discussions are provided, including omnimodal understanding (Section 3.2), vision interaction (Section 3.4), speech interaction (Section 3.3), and multimodal retrieval (Section 3.5). Finally, the ablation study (Section 3.6) is presented to examine the contributions of different components in NExT-OMNI, offering deeper insights into the factors that drive its success. For experiments on single-turn interactions such as text-to-image and text-to-audio generation, please refer to Appendix H for more details.

## 3.1 IMPLEMENTATION DETAILS

**Model Configurations.** We initialize our vision encoder and audio encoder with CLIP-ViT-Large (Dosovitskiy et al., 2021) and Whisper-Turbo (Radford et al., 2023) weights, respectively, then perform joint training for reconstruction and semantic alignment using auxiliary VQVAE (Van Den Oord et al., 2017) and corresponding decoders. Specifically, we conduct warmup training on 70M image-text pairs constructed from LAION (Schuhmann et al., 2022) and DataComp (Li et al., 2024d). The image resolution is set to 256×256 with a downsampling rate of 16. We reuse the text encoder from CLIP-ViT to extract caption semantics for the computation of the contrastive loss $\mathcal{L}_{\text{constra}}^{\text{V}}$. For the audio encoder, we conduct warmup training using a combination of open-source datasets, including LibriSpeech (Panayotov et al., 2015), WenetSpeech (Zhang et al., 2022), and AudioCaps (Kim et al., 2019), totaling 2,000 hours, supplemented by proprietary speech and music datasets amounting to 100,000 hours. During this process, we set the maximum audio clip length to 15 seconds and employ Qwen2.5-0.5B (Yang et al., 2024) for the computation of the audio caption loss $\mathcal{L}_{\text{cap}}^{\text{A}}$. We initialize NExT-OMNI with Qwen2.5-7B (Yang et al., 2024) weights, equipped with the warmup-trained vision encoder and audio encoder, along with lightweight modality heads containing nearly 128M parameters, and conduct progressive three-stage DFM training on carefully constructed omnimodal data. Throughout this process, we set the classifier-free guidance (Ho & Salimans, 2022) probability to 0.1 for multimodal generation tasks and the response padding block size to 64 for multimodal understanding tasks. We reuse the reconstruction terms $\mathcal{L}_{\text{rec}}^{\text{A}}$ and $\mathcal{L}_{\text{rec}}^{\text{V}}$ from the modal encoder warmup. These steps enhance multimodal generation, understanding, and retrieval capabilities comprehensively.

**Pre-Training (PT).** To efficiently and stably conduct flow matching modeling, we perform omnimodal joint training during the alignment pre-training stage using short audio clips within 15 seconds, single images at 256×256 resolution with 16× downsampling, and short text with a maximum context window of 2K, leveraging large amounts of low-quality data to rapidly achieve omnimodal alignment. We train on a mixture of image-text pairs and audio-text pairs. Specifically, ImageNet-1K (Deng et al., 2009), JourneyDB (Pan et al., 2023), LAION (Schuhmann et al., 2022), and FLUX (Labs, 2024) synthetic data for image generation; re-captioned image-text pairs from COYO (Byeon et al., 2022), CommonCrawl (Common Crawl, 2007), LAION (Schuhmann et al., 2022), and DataComp (Li et al., 2024d) for image understanding; LibriSpeech (Panayotov et al., 2015), WenetSpeech (Zhang et al., 2022), AudioCaps (Kim et al., 2019), and proprietary data for audio understanding and generation. To prevent degradation of the model's textual capabilities, we sample shorter pure text data from Infinity-Instruct (Li et al., 2025a) and Evol-Instruct (Xu et al., 2024), and incorporate them into the mixed training dataset.

**Continued Pre-Training (CPT).** During the CPT stage, we increase image resolution to 384×384 and introduce long text, interleaved image-text, video, and long audio data with a maximum context window of 16K. This enables the model to support natural multi-turn visual and audio interactions while possessing the capability to understand and preliminarily generate short videos. For video data, we uniformly extract 8 frames as a multi-image input. For long audio data, we decompose them into multiple chunks with a maximum length of 15 seconds. Our experiments demonstrate that this simple strategy efficiently endows the model with video and long audio understanding and generation capabilities. Beyond partial resampling of pre-training stage data, we also incorporate PixMo (Deitke et al., 2024) and LLaVA-OneVision (Li et al., 2024a) training data for image understanding; FLUX (Labs, 2024) synthetic data for image generation; MMC4-Core (Zhu et al., 2023), OmniCorpus (Li et al., 2025b), and ShareGPT4Video (Chen et al., 2024c) supporting multi-image and short video understanding; OpenVid (Nan et al., 2024) and internal video data for video generation; OpenOmni (Luo et al., 2025a) synthetic data exceeding 15 seconds and proprietary audio data for audio understanding and generation.

**Supervised Fine-Tuning (SFT).** In the SFT stage, we train the model to learn all multimodal-related instruction data, equipping it with any-to-any generation capabilities to accomplish diverse multimodal generation and understanding tasks. We collect LLaVA-OneVision (Li et al., 2024a) and PixMo (Deitke et al., 2024) instruction data for image generation; LLaVA-Video (Zhang et al., 2025c) instruction data for video understanding; OpenOmni (Luo et al., 2025a) for multi-turn audio interaction; InterSyn (Ma et al., 2025b) for multi-turn image interaction; along with BLIP3-o (Chen et al., 2025a), ShareGPT-4o-Image (Chen et al., 2025b), Nano-consistent (Ye et al., 2025b) image generation, and TIP-I2V (Wang & Yang, 2024) video generation to construct an omnimodal training

Table 1: **Comparison with existing state-of-the-art omnimodal models on omnimodal understanding benchmarks, including OmniBench, WorldSense, and AV-Odyssey.** Here, "T", "V", and "A" represent text, vision, and audio modality inputs, respectively. We mark the best performance in **bold**.

| Model | OmniBench | | | WorldSense | | | AV-Odyssey | AVG. |
|---|---|---|---|---|---|---|---|---|
| | T+V | T+A | T+A+V | A | T+A | T+A+V | T+A+V | |
| AnyGPT (Zhan et al., 2024) | 20.1 | 16.2 | 18.0 | 16.5 | 17.2 | 16.2 | - | - |
| OmniFlow (Li et al., 2025c) | 20.4 | 18.7 | 20.6 | 20.4 | 18.9 | 19.7 | | - |
| Video-SALMONN (Sun et al., 2024a) | 34.9 | 35.9 | 35.6 | - | - | - | 25.3 | - |
| UnifiedIO2-large (Lu et al., 2024) | 29.1 | 29.1 | 27.1 | 25.2 | 20.9 | 23.3 | 26.0 | 25.8 |
| UnifiedIO2-xlarge (Lu et al., 2024) | 34.8 | 31.2 | 38.0 | 23.4 | 20.7 | 24.7 | 26.3 | 28.4 |
| UnifiedIO2-xxlarge (Lu et al., 2024) | 33.5 | 32.5 | 34.0 | 25.9 | 23.7 | 25.9 | 27.2 | 29.0 |
| NExT-GPT (Wu et al., 2023) | 22.1 | 21.4 | 24.3 | 23.2 | 27.4 | 23.3 | 25.5 | 23.9 |
| OneLLM (Han et al., 2024) | 32.3 | 28.7 | 30.5 | 23.0 | 28.6 | 22.8 | 27.4 | 27.6 |
| VideoLLaMA2 (Cheng et al., 2024) | 31.7 | 26.2 | 28.9 | 23.8 | 28.5 | 25.4 | 26.8 | 27.3 |
| VITA (Fu et al., 2024) | 33.5 | 30.1 | 33.1 | 30.5 | 32.1 | 31.2 | 26.4 | 31.0 |
| VITA 1.5 (Fu et al., 2025) | 34.7 | 31.2 | 33.4 | 32.9 | 37.5 | 36.9 | 30.6 | 33.9 |
| OpenOmni (Luo et al., 2025a) | 38.3 | 36.7 | 37.4 | 34.1 | 38.9 | 37.2 | 32.8 | 36.5 |
| **NExT-OMNI** | **41.4** | **39.5** | **40.7** | **37.2** | **42.1** | **40.5** | **36.4** | **39.7** |

dataset. To further enhance the model's understanding, reasoning, and generation reasoning capabilities, we obtain 4M high-quality reasoning capability enhancement data through MMEvol (Luo et al., 2024b) sampling and filtering, and synthesize 5M reasoning-generated image data based on FLUX (Labs, 2024) for image generation improvement. Based on high-quality instruction fine-tuning data, NExT-OMNI achieves superior performance across multiple evaluation benchmarks.

## 3.2 OMNIMODAL UNDERSTANDING

To assess the omnimodal capabilities of NExT-OMNI, we conduct evaluations against current state-of-the-art omnimodal large language models (OLLMs) across three canonical benchmarks, including OmniBench (Li et al., 2024e), WorldSense (Hong et al., 2025), and AV-Odyssey (Gong et al., 2024a). As shown in Table 1, NExT-OMNI exhibits superior or comparable performance relative to advanced autoregressive-based OLLMs under various modal combination input conditions. In comparison with OpenOmni, our model achieves a 3.2 absolute average performance improvement across the three datasets. These results indicate that discrete flow matching (DFM) demonstrates potential as a viable alternative to the autoregressive (AR) paradigm for omnimodal modeling.

## 3.3 SPEECH INTERACTION

To verify NExT-OMNI's capabilities in multi-turn speech interaction, we conduct experiments on knowledge-based LLaMA Question and Web Question, covering both speech-to-text (S→T) and speech-to-speech (S→S) tasks. As shown in Table 2, under training with multi-turn speech instruction data of a similar scale, compared to AR-based speech-language models such as Stream-Omni (Zhang et al., 2025b), NExT-OMNI demonstrates competitive knowledge-based speech interaction capabilities on Spoken QA. This indicates that DFM-based omnimodal models can handle complex scenarios of multi-turn speech interaction, providing strong support for unified omnimodal generation and understanding tasks based on DFM.

Table 2: **Comparison with existing state-of-the-art unified speech-language model on multi-turn speech interaction benchmarks Spoken QA.** Here, "T" and "S" represent text and speech (belonging to the audio modality) inputs, respectively. We mark the best performance in **bold** and the second-best performance with an underline.

| Model | Llama Q. | | Web Q. | | AVG. | |
|---|---|---|---|---|---|---|
| | S→T | S→S | S→T | S→S | S→T | S→S |
| SpeechGPT (Zhang et al., 2023a) | 21.6 | - | 6.5 | - | 14.1 | - |
| Moshi (Défossez et al., 2024) | 62.3 | 21.0 | 26.6 | 9.2 | 44.5 | 15.1 |
| GLM-4-Voice (Zeng et al., 2024) | 64.7 | 50.7 | 32.2 | 15.9 | 48.5 | 33.3 |
| Freeze-Omni (Wang et al., 2024c) | 72.0 | - | 44.7 | - | 58.4 | - |
| LLaMA-Omni (Fang et al., 2024) | 67.7 | 49.0 | 33.4 | 23.7 | 50.6 | 36.4 |
| VITA-1.5 (Fu et al., 2025) | 76.7 | - | 42.7 | - | 59.7 | - |
| Stream-Omni (Zhang et al., 2025b) | 76.3 | 65.0 | 44.2 | 27.5 | 60.3 | 46.3 |
| OpenOmni (Luo et al., 2025a) | 74.6 | **67.2** | 44.5 | **28.9** | 59.6 | **48.1** |
| **NExT-OMNI** | **78.4** | 66.4 | **45.6** | 28.3 | **62.0** | 47.4 |

## 3.4 VISION INTERACTION

To explore the high-level capabilities of NExT-OMNI in multi-turn vision interaction, we evaluate it on the interleaved image-text generation benchmark OpenING (Zhou et al., 2025). This bench-

Table 3: **Comparison with existing state-of-the-art unified vision-language model on multi-turn vision interaction benchmarks OpenING.** We mark the best performance in **bold**.

| Model | GPT Evaluation | | | | IntJudge Evaluation | | | | AVG. |
|---|---|---|---|---|---|---|---|---|---|
| | FDT | w/o Tie | w/ Tie (0) | w/ Tie (.5) | FDT | w/o Tie | w/ Tie (0) | w/ Tie (.5) | |
| MiniGPT-5 (Zheng et al., 2023) | 28.6 | 28.4 | 28.0 | 28.7 | 24.5 | 15.5 | 9.9 | 27.9 | 23.9 |
| NExT-GPT (Wu et al., 2023) | 22.6 | 22.4 | 22.1 | 22.7 | 31.0 | 21.7 | 13.4 | 32.6 | 23.5 |
| DEEM (Luo et al., 2024a) | 25.6 | 25.4 | 25.2 | 25.9 | 31.2 | 21.3 | 13.6 | 32.3 | 25.1 |
| Show-o (Xie et al., 2024) | 30.8 | 30.2 | 29.6 | 30.6 | 31.5 | 21.1 | 12.5 | 32.9 | 27.4 |
| Emu2 (Sun et al., 2024b) | 41.7 | 41.6 | 40.6 | 41.9 | 36.3 | 33.8 | 21.9 | 39.5 | 37.2 |
| SEED-LLaMA (Ge et al., 2023b) | 41.0 | 40.9 | 40.5 | 41.0 | 50.1 | 47.7 | 31.6 | 48.5 | 42.7 |
| VILA-U (Wu et al., 2024c) | 50.5 | 50.1 | 50.3 | 50.5 | 51.4 | 51.2 | 32.3 | 50.9 | 48.4 |
| Anole (Chern et al., 2024) | 53.4 | 53.1 | 52.6 | 53.1 | 53.4 | 52.0 | 33.9 | 51.3 | 50.4 |
| SEED-X (Ge et al., 2024) | 54.8 | 55.1 | 54.1 | 55.0 | 49.9 | 49.6 | 33.6 | 49.7 | 50.2 |
| MMaDA (Yang et al., 2025) | 51.4 | 52.6 | 51.1 | 52.5 | 47.6 | 47.2 | 31.8 | 47.4 | 47.7 |
| FUDOKI (Wang et al., 2025) | 47.6 | 49.2 | 47.8 | 48.6 | 44.4 | 44.1 | 30.1 | 44.2 | 44.5 |
| **NExT-OMNI** | **58.7** | **58.3** | **57.4** | **58.6** | **56.3** | **57.7** | **37.5** | **55.4** | **55.0** |

mark requires models to perform multi-turn interleaved image generation based on input content and employs additional judge models for content consistency scoring, challenging the model's ability to naturally determine image generation positions and contextual understanding capabilities. As shown in Table 3, compared to vision-language unified models MMaDA (Yang et al., 2025) and FUDOKI (Wang et al., 2025) with similar architectures, NExT-OMNI demonstrates superior performance in multi-turn interactive generation for real-world usage scenarios, reflecting NExT-OMNI's advantages in general capabilities under multi-turn real-world contexts. Furthermore, compared to AR-based classical works VILA-U (Wu et al., 2024c) and SEED-X (Ge et al., 2024), NExT-OMNI also exhibits superior effectiveness, indicating that the DFM strategy possesses considerable potential in interactive generation consistency and merits further attention.

## 3.5 MULTIMODAL RETRIEVAL

To provide more insights into the impact of paradigms and unified representations in broader multimodal task scenarios such as multimodal retrieval, we adopt the MM-Embed (Lin et al., 2024) approach to sample a 100K subset from the dataset M-BEIR (Wei et al., 2024) for multimodal retrieval training. Specifically, for input multimodal queries and retrieval candidates, we extract features from the <EOS> token after model encoding for multimodal retrieval ranking fine-tuning, and test on multiple multimodal retrieval benchmarks, including InfoSeek (Chen et al., 2023), OVEN (Hu et al., 2023), FashionIQ (Wu et al., 2021), and CIRR (Liu et al., 2021). We select classical works (Wu et al., 2024a; Deng et al., 2025; Wang et al., 2025; Wu et al., 2024c; Xie et al., 2024; Yang et al., 2025) with different paradigms and representations, and report Top 5 retrieval accuracy in Table 4.

We observe two phenomena. First, models based on discrete flow or diffusion (such as FU-DOKI (Wang et al., 2025) and MMaDA (Yang et al., 2025)) outperform AR or hybrid architecture models (Wu et al., 2024a; Deng et al., 2025; Xie et al., 2024; Wu et al., 2024c; Ye et al., 2025c). We attribute this to the fact that corrective bidirectional information encoding training methods can better aggregate contextual multimodal information compared to AR architectures based on causal masking mechanisms. During single feature extraction, they degrade to BERT-like feature extraction approaches (Lee et al., 2024), providing superior multimodal representations and demonstrating broader application potential of DFM. Another finding is that while using decoupling mechanisms (multiple encoders and MOT (Liang et al., 2024) mechanisms like Bagel) performs better on multimodal understanding and generation tasks, compared to unified representation methods (Wu et al., 2024c; Xie et al., 2024; Yang et al., 2025; Li et al., 2024c; Zhang et al., 2026), they essentially involve routing between different models. The encoded features remain overly separated, making it difficult to produce unified representations, resulting in suboptimal performance on feature similarity-based multimodal retrieval tasks, which conversely limits the application scenarios of these methods. Based on these two findings, NExT-OMNI, which employs unified representation and DFM paradigm modeling, also demonstrates considerable potential in application scenarios beyond multimodal generation and understanding.

Table 4: **Comparison with existing classic unified models on various multimodal retrieval benchmarks, including InfoSeek, OVEN, FashionIQ, and CIRR.** Here, "T", "A", and "V" represent text, vision, and audio modality inputs, respectively. We mark the best performance in **bold**.

| Model | Paradigm | Rep. | InfoSeek | | OVEN | | FashionIQ | CIRR | AVG. |
|---|---|---|---|---|---|---|---|---|---|
| | | | V+T→T | V+T→V+T | V+T→T | V+T→V+T | V+T→V | V+T→V | |
| Janus (Wu et al., 2024a) | AR | Decoupled | 21.3 | 35.4 | 22.4 | 37.8 | 12.4 | 30.1 | 26.6 |
| Bagel (Deng et al., 2025) | AR+Diff. | Decoupled | 23.1 | 38.2 | 24.5 | 39.6 | 13.1 | 32.4 | 28.5 |
| FUDOKI (Wang et al., 2025) | DFM | Decoupled | 25.4 | 40.0 | 25.3 | 41.6 | 15.5 | 34.9 | 30.5 |
| VILA-U (Wu et al., 2024c) | AR | Unified | 23.3 | 37.4 | 24.0 | 38.5 | 13.6 | 33.8 | 28.4 |
| Show-o (Xie et al., 2024) | AR+Discrete Diff. | Unified | 24.8 | 39.3 | 25.6 | 42.5 | 15.9 | 35.2 | 30.6 |
| MMaDA (Yang et al., 2025) | Discrete Diff. | Unified | 25.9 | 40.8 | 26.5 | 43.7 | 17.5 | 36.3 | 31.8 |
| **NExT-OMNI** | DFM | Unified | **27.6** | **41.5** | **27.1** | **44.6** | **18.9** | **37.6** | **32.9** |

Table 5: **Ablation study on several key components of NExT-OMNI.** Here, "S" represents speech (belonging to the audio modality) inputs. We mark the best performance in **bold**.

| Paradigm | Rep. | DGS. | Recon. | Und. | | Gen. | | Retrieval | | AVG. |
|---|---|---|---|---|---|---|---|---|---|---|
| | | | | VQA$^{v2}$ | AudioCaps | GenEval | Spoken QA (S→S) | InfoSeek | OVEN | |
| AR | Decoupled | ✗ | ✗ | 55.2 | 62.8 | 53.4 | 16.4 | 28.3 | 32.1 | 41.4 |
| DFM | Decoupled | ✗ | ✗ | 52.3 | 60.1 | 59.8 | 20.3 | 29.6 | 33.7 | 42.6 |
| DFM | Unified | ✗ | ✗ | 51.7 | 59.4 | 59.2 | 19.5 | 32.8 | 35.6 | 43.0 |
| DFM | Unified | ✓ | ✗ | 54.3 | 61.9 | 59.4 | 19.8 | 33.1 | 35.4 | 43.9 |
| DFM | Unified | ✓ | ✓ | **56.2** | **63.4** | **62.6** | **21.7** | **33.7** | **36.1** | **45.6** |

## 3.6 ABLATION STUDY

We conduct ablation studies on key components of NExT-OMNI, including the modeling paradigm, representation methods, dynamic generation strategy (DGS), and reconstruction item. Benchmarks include image understanding (VQA$^{v2}$ (Goyal et al., 2017)), audio understanding (Audio-Caps (Kim et al., 2019)), image generation (GenEval (Ghosh et al., 2023)), speech generation (Spoken QA (Fang et al., 2024)), and multimodal retrieval (InfoSeek and OVEN).

Compared to the AR-based baseline with decoupled representations (i.e., an understanding-oriented encoder and a generation-oriented encoder), replacing the training paradigm with DFM (see Table 5) leads to a decline in understanding performance, but yields notable improvements in generation and retrieval tasks. When shifting further to unified representations, conflicts emerge due to the different granularity requirements of generation and understanding, resulting in an overall performance drop. Nevertheless, feature-based multimodal retrieval still benefits under this setting. These observations are consistent with our earlier findings, suggesting that unified representations may hold greater potential for broader applications.

When we introduce the DGS during training to better serve text generation tasks requiring dynamic length generation capabilities, we can significantly improve the performance on multimodal understanding tasks, achieving competitive performance with AR models. When we incorporate modality reconstruction loss terms during training, the model's performance on generation and retrieval tasks is significantly enhanced, while also providing some gains for understanding tasks. This indicates that reconstruction can add more low-level fine-grained information constraints to features encoded by visual encoders, alleviating the model's bias toward excessive focus on high-level semantic information, thereby enhancing fine-grained information in unified representations and providing good support for subsequent generation and retrieval tasks.

## 4 CONCLUSION

In this paper, we introduce NExT-OMNI, the first omnimodal foundation model fully built on discrete flow matching, which supports understanding, generation, and retrieval across text, images, video, and audio within a unified architecture. By incorporating reconstruction-feedback-enhanced unified representations and dynamic-length generation strategies, NExT-OMNI achieves deep fusion of multimodal features while substantially reducing model complexity. This design not only strengthens generation, understanding, and retrieval capabilities, but also establishes a new paradigm for unified multimodal modeling. Extensive experiments demonstrate the effectiveness of NExT-

OMNI and provide insights into how architectural design interacts with unified representations in multimodal tasks. Looking ahead, we plan to extend NExT-OMNI to broader domains, such as action trajectory generation in vision-language-action models and video generation for physical AI understanding in world models, where we expect it to play an even greater role.

## ACKNOWLEDGEMENTS

Min Yang was supported by National Key Research and Development Program of China (2024YFF0908200), National Natural Science Foundation of China (Grant No. 62376262), and Natural Science Foundation of Guangdong Province of China (2024A1515030166, 2025B1515020032).

This research/project is supported by the National Research Foundation, Singapore under its National Large Language Models Funding Initiative (AISG Award No: AISG-NMLP2024-002). Any opinions, findings and conclusions or recommendations expressed in this material are those of the author(s) and do not reflect the views of National Research Foundation, Singapore.

## ETHICS STATEMENT

This work advances unified omnimodal large language models by introducing discrete flow matching modeling paradigms and reconstruction-enhanced unified representations, enhancing multimodal understanding, generation, and retrieval capabilities, enabling efficient text-vision-audio integration for applications such as assistive tools, creative content, and education. However, high-quality image and audio generation also poses risks, including potential misuse for misinformation or manipulation. While our model is not identity-specific, downstream applications should include safeguards such as watermarking and prompt filtering. We advocate for ethical use, emphasizing fairness, robustness, and transparency.

## REPRODUCIBILITY STATEMENT

Experimental settings are carefully described and listed in Section 3. We detail the model design, data curation, and supplement implementation details in Appendix D, Appendix E, and Appendix F, respectively. To further ensure reproducibility, we release training details, data protocols, and open-source both the code and model checkpoints.

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

# Appendix

## A  Preliminaries of Discrete Flow Matching

We provide a brief introduction to the mathematical processes of training and inference in discrete flow matching (DFM) (Shaul et al., 2024). Analogous to continuous flow matching (Lipman et al., 2022), DFM defines a family of time-dependent probability distributions $\{p_t(\mathrm{x})\}_{t \in [0,1]}$ that define a smooth transition, or probability paths, from a source distribution $p(\mathrm{x})$ to a target distribution $q(\mathrm{x})$. Here, $\mathrm{x} = \{\mathrm{x}^1, \mathrm{x}^2, \ldots, \mathrm{x}^D\}$ lies in the discrete space $\mathcal{S} = \mathcal{T}^D$, where $D$ denotes the number of discrete variables and $\mathcal{T} = [K] = \{1, 2, \ldots, K\}$ represents the finite set of possible discrete values. Each $p_t(\mathrm{x})$ is constructed as $p_t(\mathrm{x}) = \sum_{\mathrm{x}_1 \in \mathcal{S}} p_t(\mathrm{x}|\mathrm{x}_1)q(\mathrm{x}_1)$, where the conditional distribution is factorized across dimensions. Namely, $p_t(\mathrm{x}|\mathrm{x}_1) = \prod_{i=1}^{D} p_t(\mathrm{x}^i|\mathrm{x}_1^i)$. Note that each $p_t(\mathrm{x}^i|\mathrm{x}_1^i)$ defines a univariate interpolation between a base distribution $p(\mathrm{x}^i)$ and a one-hot distribution $\delta_{\mathrm{x}_1^i}(\mathrm{x}^i)$[1]. A common design for the interpolation is the mixture path, defined via a time-dependent scheduler $\kappa_t(\mathrm{x}_1^i) \in [0, 1]$, i.e.,

$$p_t(\mathrm{x}^i|\mathrm{x}_1^i) = (1 - \kappa_t(\mathrm{x}_1^i))p(\mathrm{x}^i) + \kappa_t(\mathrm{x}_1^i)\delta_{\mathrm{x}_1^i}(\mathrm{x}^i), \tag{4}$$

where $\kappa_0(\cdot) = 0$ and $\kappa_1(\cdot) = 1$. To obtain a more semantically meaningful transition path, we can adopt a metric-induced probability path as follows:

$$p_t(\mathrm{x}^i|\mathrm{x}_1^i) = \mathrm{Softmax}(-\beta_t d(\mathrm{x}^i, \mathrm{x}_1^i)), \tag{5}$$

where $d(\cdot, \cdot)$ is the cosine distance function between token embeddings, and $\beta_t \in [0, \infty]$ is a monotonic schedule.

After minimizing the kinetic energy (Shaul et al., 2024), the probability velocities can be formulated as follows:

$$u_t^i(\mathrm{x}^i, z|\mathrm{x}_1) = p_t(\mathrm{x}^i|\mathrm{x}_1^i)\frac{\partial \beta_t}{\partial t} \max\{d(z^i, \mathrm{x}_1^i) - d(\mathrm{x}^i, \mathrm{x}_1^i), 0\}. \tag{6}$$

Intuitively, for the $i$-th coordinate $z^i \in \mathcal{T}$, this velocity ensures that probability mass flows from the state $z^i$ to the state $\mathrm{x}^i$ only when $\mathrm{x}^i$ lies closer to $\mathrm{x}_1^i$ than $z^i$ does, i.e., $d(\mathrm{x}^i, \mathrm{x}_1^i) < d(z^i, \mathrm{x}_1^i)$. As a result, the flow monotonically progresses toward $\mathrm{x}_1^i$.

During training, we sample $\mathrm{x}_t$ from $p_t(\mathrm{x}^i|\mathrm{x}_1^i)$ and feed it into the model to fit the target $\mathrm{x}_1$. During inference, we employ an Euler solver for enhanced sampling robustness as recommended in (Shaul et al., 2024). This solver simulates the continuous-time Markov chain (CTMC) process $\{\mathrm{x}_t\}_{t \in [0,1]}$. Given that $\mathrm{x}_t \sim p_t$, the solver updates the $i$-th coordinate from time $t$ to $t+h$ following the procedure below:

- Sample $\mathrm{x}_1^i \sim p_{1|t}^i(\cdot|\mathrm{x}_t)$ from our model;

- Compute the total conditional transition rate $\lambda^i = \sum_{\mathrm{x}^i \neq \mathrm{x}_t^i} u_t^i(\mathrm{x}^i, \mathrm{x}_t^i|\mathrm{x}_1^i)$ (see Eq. (6));

- Draw a uniform random variable $Z_{\mathrm{change}}^i \sim \mathcal{U}[0, 1]$;

- Sample $\mathrm{x}_{t+h}^i$ as follows: if $Z_{\mathrm{change}}^i \leq 1 - \exp\{-h\lambda^i\}$, sample $\mathrm{x}_{t+h}^i$ from $\frac{u_t^i(\cdot, \mathrm{x}_t^i|\mathrm{x}_1^i)}{\lambda^i}(1 - \delta_{\mathrm{x}_t^i}(\cdot))$; otherwise set $\mathrm{x}_{t+h}^i = \mathrm{x}_t^i$.

The procedure begins with completely corrupted sequences and iteratively denoises entire sequences in parallel, enabling richer bidirectional information integration to enhance final performance. We set $\beta_t = c\left(\frac{t}{1-t}\right)^a$ with $c = 3$ and $a = 0.9$, as suggested in (Shaul et al., 2024).

## B  Related Work

**Unified Vision-Language Models.** Unified vision-language models (Team, 2024; Dong et al., 2024; Tian et al., 2024; Luo et al., 2024a; Wu et al., 2024b) have attracted significant research attention due to their powerful multimodal understanding and generation capabilities. SEED and the Emu series (Ge et al., 2023b; 2024; Sun et al., 2023; 2024b; Wang et al., 2024b) adopt discrete autoregressive modeling approaches, unifying multimodal understanding and generation through next-token

---

[1]If $\mathrm{x}^i = \mathrm{x}_1^i$, $\delta_{\mathrm{x}_1^i}(\mathrm{x}^i)=1$; else, $\delta_{\mathrm{x}_1^i}(\mathrm{x}^i)=0$.

prediction paradigms. However, this coarse unification approach is susceptible to granularity mismatches between understanding and generation tasks, hindering further performance improvements. The Janus series models (Wu et al., 2024a; Chen et al., 2025c) decouple visual encoding for understanding and generation to address granularity mismatch issues, but introducing additional encoders limits flexibility. Subsequently, VILA-U (Wu et al., 2024c) and UniTok (Ma et al., 2025a) focus on constructing unified representations, using unified encoders to alleviate granularity conflicts between generation and understanding tasks, but still adhere to autoregressive modeling paradigms and lag behind specialized models (Rombach et al., 2022a; Xie et al., 2025) in generation tasks. Some works, such as BLIP-3o (Chen et al., 2025a), introduce additional diffusion-based specialized generation models for generation optimization to achieve impressive results, but further increase methodological complexity. To balance the effectiveness of generation and understanding tasks, some works attempt to use hybrid architectures. For instance, Show-o (Xie et al., 2024) and Transfusion (Zhou et al., 2024) integrate diffusion objectives into large language models for image generation, but this design breaks the autoregressive paradigm and complicates the unification of both tasks. Bagel (Deng et al., 2025) introduces MOT (Liang et al., 2024) architecture to successfully achieve excellent performance in both tasks within a relatively unified hybrid architecture, but the introduction of these decoupling mechanisms makes generation and understanding separate components, departing significantly from the ideal of a concise unified architecture. Some works (Yang et al., 2025; Wang et al., 2025) attempt to completely abandon autoregressive architectures, pursuing unified vision-language modeling from the perspective of discrete diffusion or flow matching, achieving considerable results. However, due to the lack of robust language foundation model support and engineering optimization, speed and effectiveness remain suboptimal.

**Omnimodal Language Models.** With the advancement of multimodal research, models are increasingly shifting toward unified frameworks that seamlessly integrate diverse input and output modalities. By tokenizing different data types into shared representations, models such as AnyGPT (Zhan et al., 2024) and Unified-IO2 (Lu et al., 2024) achieve seamless cross-modal task adaptability, enabling them to process audio, text, and images without requiring significant architectural modifications. OneLLM (Han et al., 2024) and NExT-GPT (Wu et al., 2023) enhance generation and understanding performance by unifying input spaces and introducing additional diffusion heads. Meanwhile, video-SALMONN (Sun et al., 2024a) enhances video understanding by incorporating fine-grained temporal modeling, improving the model's ability to interpret speech and actions within videos. To enhance human-computer interaction, the VITA series (Fu et al., 2024; 2025) introduces duplex communication schemes, enabling fluid and intuitive exchanges between users and AI models. EMOVA (Chen et al., 2024b) and OpenOmni series (Luo et al., 2024b; 2025a; Zhang et al., 2025a) further extend the expressive capabilities of multimodal systems by integrating controllable emotional speech synthesis, providing more natural and engaging user interactions. The Qwen-Omni series (Xu et al., 2025a;b) further expanded the scale of training models and data, greatly enhancing the performance of the omni models. However, these works still rely on autoregressive architectures and additional large-scale continuous flow matching modeling heads for omnimodal modeling, while more unified and lightweight discrete flow matching and diffusion architectures remain unexplored. To address this gap, we propose NExT-OMNI in this work.

## C  LIMITATION AND DISCUSSION

**Limitation.** Due to resource constraints, we conduct training and validation only at the 7B parameter scale with 2T tokens. While NExT-OMNI provides insights into how discrete flow matching can better unify generation, understanding, and retrieval tasks, its full potential has not been demonstrated due to the lack of corresponding large language model foundation support. In the future, we hope to explore broader application scenarios, such as trajectory generation in vision-language-action models and world model exploration, where visual generation assists physical perception, to demonstrate the potential of NExT-OMNI.

**Discussion.** Here, we provide further discussion on the future of omnimodal unified models. Some argue that building unified models consumes substantial resources yet struggles to achieve performance comparable to generation-only and understanding-only models, questioning the necessity of constructing unified models. We address this concern as follows: unified models are built to achieve greater generalizability. In the future, unified omnimodal models will serve as a "world brain" to interact with the real world, with their general capabilities expected to be continuously

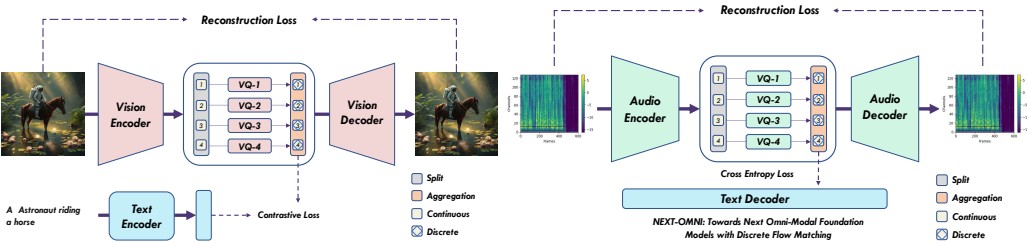

Figure 4: **The Pipeline of vision encoder (left) and audio encoder (right) warmup training.** During the modal encoder warmup training phase, self-supervised reconstruction is performed through additional quantizers and modal decoders, while semantic alignment at different granularities is achieved through text encoders or text decoders.

enhanced through multimodal data collected via interactions, thereby evolving iteratively to expand the model's capability boundaries and ultimately achieve Artificial General Intelligence (AGI). Our goal in building unified models is to enable mutual reinforcement among diverse tasks such as understanding and generation, ultimately expanding the boundaries of intelligence. For instance, through extensive vision generation learning, models can deepen their understanding of physical laws by precisely generating imagined moments, subsequently producing reasonable actions. Conversely, a deeper understanding of physical laws can enable models to generate more realistic image sequences. Based on this rationale, while building unified models may incur certain performance trade-offs, their general capabilities offer more promising prospects. On this spiraling path of development with its inevitable challenges, we must persist in this direction.

## D    MODEL DESIGN

Here, we provide additional details regarding the design principles and training process of modality encoders and modality heads.

### D.1    MODALITY ENCODERS

Similar to previous work (Luo et al., 2024a; Wu et al., 2024c; Ma et al., 2025a; Ye et al., 2025d; Wan et al., 2025; Luo et al., 2025b; Lv et al., 2026; Zhou et al., 2026; Zheng et al., 2025; Liu et al., 2025a), we design modality encoders based on unified representation methods, conducting simultaneous self-supervised reconstruction and semantic alignment optimization during the warmup training stage. As shown in Figure 4, we employ additional quantizers and modality decoders to assist reconstruction training, while utilizing text encoders and decoders for sentence-level and token-level semantic alignment training, respectively. To pursue superior performance, we adopt multi-codebook quantization (MCQ) (Ma et al., 2025a) for quantization, where separating the codebook into multiple independent sub-codebooks significantly enhances both reconstruction and semantic alignment effectiveness. However, we also observe that while increasing the number of sub-codebooks improves performance on reconstruction and downstream multimodal understanding tasks, it degrades performance on downstream multimodal generation tasks, as predicting multiple sub-codebook indices at a single position becomes more challenging. Therefore, to achieve an optimal trade-off, we set vocabulary sizes of $4\times4096$ and $2\times2048$ for the vision encoder and audio encoder in the warmup training, respectively.

**Vision Encoder.** We initialize our vision encoder with CLIP-ViT-Large (Dosovitskiy et al., 2021) weights and conduct unified representation training on nearly 70M image-text pairs composed of LAION (Schuhmann et al., 2022) and DataComp (Li et al., 2024d), incorporating both sentence-level image-text contrastive loss $\mathcal{L}_{\text{constra}}^{\text{V}}$ and VQVAE-based reconstruction loss $\mathcal{L}_{\text{rec}}^{\text{V}}$ optimization. In more detail, the VQVAE-based reconstruction loss consists of a pixel-level reconstruction loss $\mathcal{L}_{\text{R}}^{\text{V}}$, a perceptual loss $\mathcal{L}_{\text{P}}^{\text{V}}$ based on the LPIPS metric (Zhang et al., 2018), a discriminator loss $\mathcal{L}_{\text{G}}^{\text{V}}$ to enhance reconstruction fidelity (Karras et al., 2019), and a vector quantization loss $\mathcal{L}_{\text{VQ}}^{\text{V}}$ to minimize distance between the encoder output and its nearest code entry. It is denoted as $\mathcal{L}_{\text{rec}}^{\text{V}} =$

Table 6: **Comparison with existing state-of-the-art vision tokenizer and audio tokenizer.** We mark the best performance in **bold**.

| Model | Codebooks | UTMOS↑ | PESQ↑ | STOI↑ | Model | Codebooks | rFID↓ | Acc↑ |
|---|---|---|---|---|---|---|---|---|
| WavTokenizer | 4096 | 4.048 | 2.373 | 0.914 | UniTok | $8\times4096$ | 0.38 | 78.6 |
| Audio Encoder (Ours) | $2\times2048$ | **4.126** | **2.467** | **0.923** | Vision Encoder (Ours) | $4\times4096$ | **0.33** | **79.4** |

$\mathcal{L}_R^V + \lambda_{VQ} \cdot \mathcal{L}_{VQ}^V + \lambda_P \cdot \mathcal{L}_P^V + \lambda_G \cdot \mathcal{L}_G^V$, where $\lambda$ denotes the weight factor for the corresponding loss term. The image-text contrastive loss term $\mathcal{L}_{constra}^V$ is basically the same as in CLIP-VIT (Dosovitskiy et al., 2021). Therefore, the final loss term can be written as $\mathcal{L}_{overall}^V = \mathcal{L}_{rec}^V + \mathcal{L}_{contra}^V$. During this process, we maintain training hyperparameters consistent with UniTok (Ma et al., 2025a).

**Audio Encoder.** We initialize our audio encoder with Whisper-Turbo (Radford et al., 2023) weights and conduct unified representation training on nearly 102K hours of audio-text pairs composed of LibriSpeech (Panayotov et al., 2015), WeNetSpeech (Zhang et al., 2022), AudioCaps (Kim et al., 2019), and proprietary data, incorporating both token-level audio caption loss $\mathcal{L}_{cap}^A$ and VQVAE-based reconstruction loss $\mathcal{L}_{rec}^A$ optimization. The VQVAE-based reconstruction loss consists of a mel-spectrum reconstruction loss $\mathcal{L}_R^A$, a feature matching loss $\mathcal{L}_F^A$ based on a L2 norm loss, a discriminator loss $\mathcal{L}_G^A$ (Ji et al., 2024), and a vector quantization loss $\mathcal{L}_{VQ}^A$ to minimize distance between the encoder output and its nearest code entry. It is denoted as $\mathcal{L}_{rec}^A = \mathcal{L}_R^A + \lambda_{VQ} \cdot \mathcal{L}_{VQ}^A + \lambda_F \cdot \mathcal{L}_F^A + \lambda_G \cdot \mathcal{L}_G^A$, where $\lambda$ is the weight factor for the corresponding loss term. The audio-text caption loss term $\mathcal{L}_{cap}^A$ is basically the same as in Qwen-Audio (Chu et al., 2023). Therefore, the final loss term can be written as $\mathcal{L}_{overall}^A = \mathcal{L}_{rec}^A + \mathcal{L}_{cap}^A$. During this process, we maintain training hyperparameters consistent with WavTokenizer (Ji et al., 2024).

**Quantitative and Qualitative Analysis.** To validate the effectiveness of our warmup-trained vision encoder, we conduct image reconstruction and classification evaluation on ImageNet (Deng et al., 2009), reporting rFID (IDEFICS, 2023) and zero-shot classification accuracy, and compare with the state-of-the-art unified representation vision encoder UniTok (Ma et al., 2025a). Similar to the vision encoder configuration, we perform speech reconstruction comparison tests on the LibriSpeech test-clean split (Panayotov et al., 2015), employing UTMOS (Saeki et al., 2022), PESQ (Rix et al., 2001), and STOI (Ji et al., 2024) metrics, and compare with the state-of-the-art WavTokenizer (Ji et al., 2024). As shown in Table 6, our modality encoders demonstrate superior reconstruction and semantic classification performance, capable of producing reliable unified representations for multimodal generation, understanding, and retrieval. Furthermore, to more intuitively demonstrate the superiority of our modal encoders, we provide visualization results of reconstructions. As shown in Figure 6, we can observe that our vision encoder achieves better reconstruction performance compared to UniTokin details such as font edges and bear eye colors. As shown in Figure 5, we can see that our audio encoder produces clearer reconstructed mel-spectrograms for different audio types compared to WavTokenizer. These results validate the effectiveness of our modality encoder warmup training.

Table 7: **Ablation study on the scalability of NExT-OMNI.** Here, "S" represents speech (belonging to the audio modality) inputs. We mark the best performance in **bold**.

| Model Size | Training Steps | Und. | | | Gen. | | | AVG. |
|---|---|---|---|---|---|---|---|---|
| | | VQA$^{v2}$ | AudioCaps | ActivityNet-QA | GenEval | Spoken QA (S→S) | VBench | |
| 0.5B | 0.5K | 42.1 | 48.6 | 28.3 | 18.2 | 8.4 | 12.7 | 26.4 |
| 1.5B | 0.5K | 45.3 | 52.3 | 31.7 | 22.8 | 11.2 | 16.3 | 30.0 |
| 7B | 0.5K | 48.7 | 55.9 | 34.2 | 28.5 | 13.7 | 20.1 | 33.5 |
| 7B | 1K | 52.4 | 59.1 | 36.1 | 45.2 | 17.3 | 27.4 | 39.6 |
| 7B | 1.5K | 54.9 | 61.7 | 37.4 | 58.9 | 20.1 | 32.8 | 44.3 |
| 7B | 2K | **56.2** | **63.4** | **37.8** | **62.6** | **21.7** | **34.6** | **46.1** |

## D.2 MODALITY HEADS

Since we employ multi-codebook quantization (MCQ) for quantization, prediction requires forecasting multiple sub-codebook indices for a single position, which poses challenges for vision and

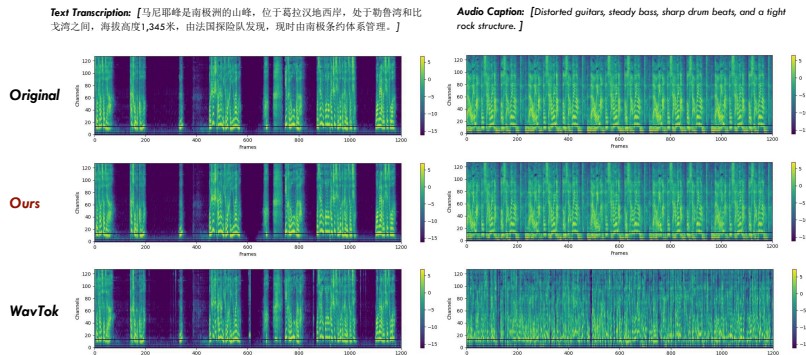

Figure 5: **Qualitative results on audio reconstruction in a max duration of 15s.**

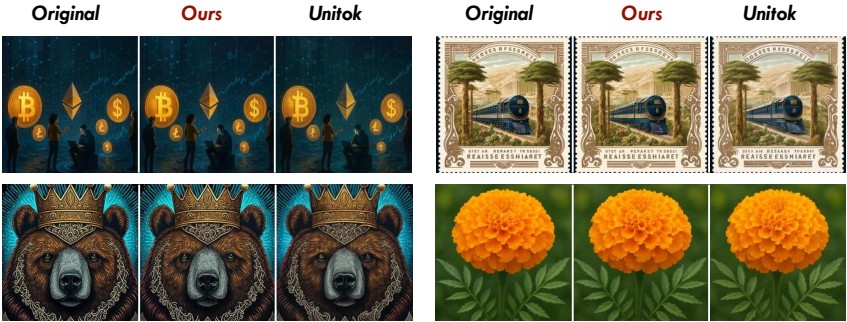

Figure 6: **Qualitative results on image reconstruction in a resolution of 256×256.**

audio head architectures. To overcome this problem, we design two different modality head structures, as illustrated in Figure 7. The left structure represents an autoregressive multi-sub-codebook index prediction modal head, which uses the hidden features output by NExT-OMNI at each position as conditions to complete the prediction of multiple sub-codebook indices through the next-token prediction paradigm. The right structure employs multiple separate heads to complete the prediction of multiple sub-codebook indices in parallel through multi-token prediction (Gloeckle et al., 2024; Liu et al., 2025b). Under equal parameter counts, the former provides more stable prediction results compared to the latter at the cost of slightly increased computational overhead, and is therefore adopted in our framework.

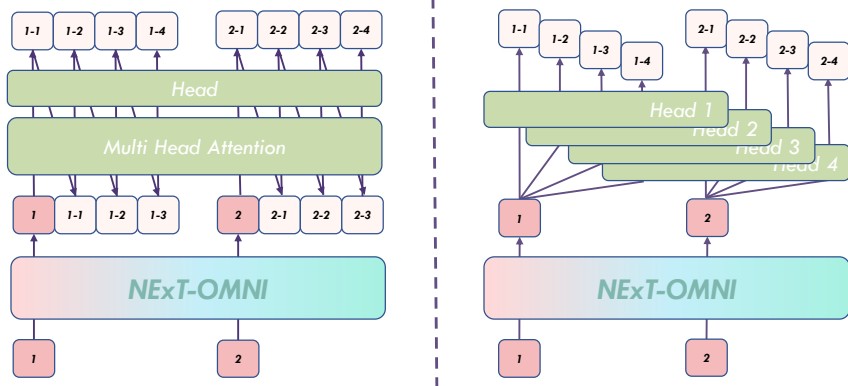

Figure 7: **Illustration of modality heads.** We design two different lightweight modal head structures for multiple sub-codebook indices prediction: next-token prediction (left) and multi-token prediction (right).

Table 8: **Overview of stages, training data, tasks, and sizes used for building NExT-OMNI.** Here, "I", "T", "A", and "V" are the abbreviations of the image, text, audio, and vision, respectively.

| Model | Stage | Data | Task | Size |
|---|---|---|---|---|
| Vision Encoder | Warmup | LAION (Schuhmann et al., 2022), DataComp (Li et al., 2024d),COYO (Byeon et al., 2022) | I→I, I→T | 70M |
| Audio Encoder | Warmup | LibriSpeech (Panayotov et al., 2015), WenetSpeech (Zhang et al., 2022) AudioCaps (Kim et al., 2019), Proprietary Data | A→A, A→T | 102K Hours |
| NExT-OMNI | PT | LLaVA-Recap-CC12M (Li et al., 2024a; Common Crawl, 2007), DataComp (Li et al., 2024d) | I→T | 32M |
| | | Blip3-o (Chen et al., 2025a), ImageNet-1K (Deng et al., 2009), JourneyDB (Pan et al., 2023) | T→I | 25M |
| | | Infinity-Instruct (Li et al., 2025a), Evol-Instruct (Xu et al., 2024) | Text | 4M |
| | | Proprietary Data, AudioCaps (Kim et al., 2019) LibriSpeech (Panayotov et al., 2015), WenetSpeech (Zhang et al., 2022) | A→T | 16M |
| | | Proprietary Data, AudioCaps (Kim et al., 2019) | T→A | 6M |
| | CPT | LLaVA-O (Li et al., 2024a), PixMo (Deitke et al., 2024),MAmmoTH-VL (Guo et al., 2024) | I→T | 10M |
| | | Blip3-o (Chen et al., 2025a), Flux-Reason (Fang et al., 2025) | T→I | 12M |
| | | MMC4-core (Zhu et al., 2023), OmniCorpus (Li et al., 2025b) ShareGPT4Video (Chen et al., 2024c) | V→T | 12M |
| | | OpenVid (Nan et al., 2024), Internal Video Data | T→V | 2M |
| | | Infinity-Instruct (Li et al., 2025a), Evol-Instruct (Xu et al., 2024) | Text | 2.3M |
| | | Proprietary Data, AudioCaps (Kim et al., 2019) | A→T | 12M |
| | | OpenOmni (Luo et al., 2025a), AudioCaps (Kim et al., 2019) | T→A | 1.3M |
| | SFT | LLaVA-O (Li et al., 2024a), PixMo (Deitke et al., 2024), MMEvol (Luo et al., 2024b), Und-4M | I→T | 7.6M |
| | | OpenOmni (Luo et al., 2025a) | T+A→T+A | 0.5M |
| | | InterSyn (Ma et al., 2025b) | T+I→T+I | 1.7M |
| | | Infinity-Instruct (Li et al., 2025a), Evol-Instruct (Xu et al., 2024) | Text | 0.8M |
| | | LLaVA-Video (Zhang et al., 2025c) | V→T | 0.9M |
| | | OpenVid (Nan et al., 2024), TIP-I2V (Wang & Yang, 2024) | T→V,T+I→V | 1M |
| | | BLIP3-o-60K (Chen et al., 2025a), ShareGPT-4o-Image (Chen et al., 2025b) Gen-5M, Nano-consistent (Ye et al., 2025b) | T→I, T+I→I | 5.8M |

# E   DATA CURATION

## E.1   DATASET CURATION DETAILS

In addition to proprietary data, we collect publicly available data encompassing diverse tasks for three-stage progressive training, including generation-related tasks such as text-to-image, text-to-video, and text-to-audio, as well as understanding-related tasks, including image understanding, video understanding, and audio understanding. We summarize the important information of all training data in Table 8.

## E.2   DATASET SYNTHETIC DETAILS

**Visual Understanding**. We randomly sample 1.2M data from LLaVA-OneVision (Li et al., 2024a) and PixMo (Deitke et al., 2024) datasets as seed data, and apply the MMEvol (Luo et al., 2024b) algorithm using the Qwen2.5-VL (Bai et al., 2025) model as the generator. Through three rounds of evolution, we construct approximately 4M instruction data (Und-4M) that are rich in diversity and complexity, to enhance the model's capability for solving complex problems.

**Image Generation**. Based on user prompts from JourneyDB (Pan et al., 2023) and Midjourney-Prompts datasets, we use the Qwen2.5 (Yang et al., 2024) model to construct numerous similar synthetic prompts containing more fine-grained descriptions. Subsequently, we employ FLUX (Labs, 2024) as the image generator to rapidly generate large quantities of images based on user and synthetic prompts under the configuration of 4 sampling steps and $512\times512$ resolution. Finally, we collect approximately 5M high-quality synthetic data (Gen-5M).

# F   ADDITIONAL IMPLEMENTATION DETAILS

We report additional implementation details about training hyper-parameters and training data in Table 9.

Table 9: **Training recipes for NExT-OMNI.** The three training stages are introduced in Section 3.1. `Stage I`: Pre-Training (PT), `Stage II`: Continue Pre-Training (CPT), `Stage III`: Supervised Fine-tuning (SFT).

| | Stage I | Stage II | Stage III |
|---|---|---|---|
| Phase | PT | CPT | SFT |
| *Training Hyper-Parameters* | | | |
| Image Resolution | 256×256 | 384×384 | 384×384 |
| Audio Duration | ≤15s | ≥15s | ≥15s |
| Video Frames | | ≤8 | ≤8 |
| LLM | Qwen2.5 7B | Qwen2.5 7B | Qwen2.5 7B |
| Learning Rate | 2e-5 (modality encoder&decoder) 1e-4 (others) | 1e-6 (modality encoder&decoder) 2e-5 (others) | 1e-6 (modality encoder&decoder) 2e-5 (others) |
| Optimizer | AdamW | AdamW | AdamW |
| Optimizer hyper-parameters | $\beta_1, \beta_2, \epsilon$=0.9, 0.995, 1e-6 | $\beta_1, \beta_2, \epsilon$=0.9, 0.999, 1e-8 | $\beta_1, \beta_2, \epsilon$=0.9, 0.999, 1e-8 |
| Weight Decay | 0.05 | 0.05 | 0.05 |
| Training iterations | 10K | 18K | 25K |
| Warmup steps | 1K | 500 | 500 |
| Learning Rate Scheduler | Cosine | Cosine | Cosine |
| Batch Size Per GPU | 16 | 8 | 4 |
| Maximum Token Length | 2K | 16K | 16K |
| *Training Data* | | | |
| Data Size | ∼83M | ∼52M | ∼18M |
| Data Type | Pair | Pair/Interleave | Instruction |

## G ADDITIONAL ABLATION STUDY

Although NExT-OMNI demonstrates competitive performance against the SOTA methods, its scalability has yet to be validated. As is well known, scalability is crucial for model performance. We conduct ablation experiments to assess the scalability concerning data count and model size. As shown in Table 7, gradually increasing the training data enables the model to successfully scale while achieving improved results. Additionally, increasing the sizes of backbone leads to sustained performance enhancements, indicating that NExT-OMNI possesses good scalability.

## H ADDITIONAL EXPERIMENTS RESULTS

**Image Understanding.** We evaluate the image understanding capabilities of NExT-OMNI on several benchmarks, including POPE (Li et al., 2023b), MME-P (Fu et al., 2023), SEED (Ge et al., 2023a), MMB (Liu et al., 2024c), GQA (Ainslie et al., 2023), MMMU (Yue et al., 2024), and MM-Vet (Yu et al., 2024). As shown in Table 3, NExT-OMNI not only outperforms models MMaDA and FUDOKI with similar architectures, but also achieves highly competitive results compared to autoregressive (AR)-based multimodal large language models. Notably, NExT-OMNI unleashes the potential of discrete flow matching (DFM) in understanding tasks through dynamic length generation and caching methods, while achieving faster generation speeds compared to AR models as shown in Figure 12. These results demonstrate that DFM represents a highly promising alternative to AR architectures and merits significant attention.

**Image Generation.** We evaluate the image generation capabilities of NExT-OMNI on the widely used GenEval (Ghosh et al., 2023) and DPG-Bench (Hu et al., 2024) benchmarks. NExT-OMNI achieves competitive overall performance in both Table 11 and Table 12, scoring 0.85 on GenEval and 84.46 on DPG-Bench, demonstrating strong performance in both generation-only and understanding-generation categories with smaller model parameters and faster response speeds. These results highlight the advantages of the discrete flow matching framework modeling, which allows bidirectional integration of visual information, first generating image layouts and then progressively filling in details, better aligning with the natural characteristics of image generation.

**Audio Generation and Understanding.** We validate our audio understanding and generation capabilities on benchmarks LibirSpeech (Panayotov et al., 2015) and AudioCaps (Kim et al., 2019). As shown in Table 13, compared to other omnimodal models, NExT-OMNI, as the first model to employ discrete flow matching modeling, achieves significantly superior performance in natural audio generation, understanding, speech translation, and speech synthesis. These results further validate the generalizability and scalability of discrete flow matching modeling, providing potential for future applications in protein molecular structure prediction, 3D model generation, and other domains.

Table 10: **Multimodal understanding performance on various benchmarks.** "Und." and "Gen." denote the abbreviations of "Understanding" and "Generation". [†] denotes models that integrate an external pre-trained diffusion model.

| Model | Paradigm | # Params | POPE | MME-P | MMB | SEED | GQA | MMMU | MM-Vet |
|---|---|---|---|---|---|---|---|---|---|
| **Und. Only** | | | | | | | | | |
| LLaVA (Liu et al., 2024b) | AR | 7B | 76.3 | 809.6 | 38.7 | 33.5 | - | - | 25.5 |
| LLaVA-v1.5 (Liu et al., 2023) | AR | 7B | 85.9 | 1510.7 | 64.3 | 58.6 | 62.0 | 35.4 | 31.1 |
| InstructBLIP (Dai et al., 2023) | AR | 7B | - | - | 36.0 | 53.4 | 49.2 | - | 26.2 |
| Qwen-VL-Chat (Bai et al., 2023) | AR | 7B | - | 1487.5 | 60.6 | 58.2 | 57.5 | - | - |
| IDEFICS-9B (IDEFICS, 2023) | AR | 8B | - | - | 48.2 | - | 38.4 | - | - |
| Emu3-Chat (Wang et al., 2024b) | AR | 8B | 85.2 | 1244.0 | 58.5 | 68.2 | 60.3 | 31.6 | 37.2 |
| InstructBLIP (Dai et al., 2023) | AR | 13B | 78.9 | 1212.8 | - | - | 49.5 | - | 25.6 |
| **Und. and Gen.** | | | | | | | | | |
| LaVIT[†] (Jin et al., 2024) | AR | 7B | - | - | - | - | 46.8 | - | - |
| MetaMorph[†] (Tong et al., 2024) | AR | 8B | - | - | 75.2 | 71.8 | - | - | - |
| Gemini-Nano-1 (Team et al., 2023) | - | 1.8B | - | - | - | - | - | 26.3 | - |
| ILLUME (Wang et al., 2024a) | AR | 7B | 88.5 | 1445.3 | 65.1 | 72.9 | - | 38.2 | 37.0 |
| TokenFlow-XL (Qu et al., 2024) | AR | 13B | 86.8 | 1545.9 | 68.9 | 68.7 | 62.7 | 38.7 | 40.7 |
| LWM (Liu et al., 2024a) | AR | 7B | 75.2 | - | - | - | 44.8 | - | 9.6 |
| VILA-U (Wu et al., 2024c) | AR | 7B | 85.8 | 1401.8 | - | 59.0 | 60.8 | - | 33.5 |
| Chameleon (Team, 2024) | AR | 7B | - | - | - | - | - | 22.4 | 8.3 |
| Janus (Wu et al., 2024a) | AR | 1.5B | 87.0 | 1338.0 | 69.4 | 63.7 | 59.1 | 30.5 | 34.3 |
| Janus-Pro (Chen et al., 2025c) | AR | 1.5B | 86.2 | 1444.0 | 75.5 | 68.3 | 59.3 | 36.3 | 39.8 |
| Show-o (Xie et al., 2024) | AR+Discrete Diff. | 1.3B | 73.8 | 948.4 | - | - | 48.7 | 25.1 | - |
| D-Dit (Li et al., 2024f) | Discrete Diff.+Diff. | 2.0B | 84.0 | 1124.7 | - | - | 59.2 | - | - |
| FUDOKI (Wang et al., 2025) | DFM | 1.5B | 86.1 | 1485.4 | 73.9 | 68.2 | 57.6 | 34.3 | 38.0 |
| MMaDA (Yang et al., 2025) | Discrete Diff. | 8B | 86.1 | 1410.7 | 68.5 | 64.2 | 61.3 | 30.2 | - |
| NExT-OMNI | DFM | 7B | 87.4 | 1537.8 | 78.9 | 76.3 | 62.7 | 43.7 | 40.1 |

Table 11: **Visual Generation Results on GenEval (Ghosh et al., 2023).** "Und." and "Gen." denote the abbreviations of "Understanding" and "Generation".

| Model | Paradigm | Single Obj. | Two Obj. | Counting | Colors | Position | Color Attri. | Overall↑ |
|---|---|---|---|---|---|---|---|---|
| **Gen. Only** | | | | | | | | |
| SDv1.5 (Rombach et al., 2022a) | Diff. | 0.97 | 0.38 | 0.35 | 0.76 | 0.04 | 0.06 | 0.43 |
| PixArt-α (Chen et al., 2024a) | Diff. | 0.98 | 0.50 | 0.44 | 0.80 | 0.08 | 0.07 | 0.48 |
| SDv2.1 (Rombach et al., 2022a) | Diff. | 0.98 | 0.51 | 0.44 | 0.85 | 0.07 | 0.17 | 0.50 |
| Emu3-Gen (Wang et al., 2024b) | AR | 0.98 | 0.71 | 0.34 | 0.81 | 0.17 | 0.21 | 0.54 |
| SDXL (Podell et al., 2023) | Diff. | 0.98 | 0.74 | 0.39 | 0.85 | 0.15 | 0.23 | 0.55 |
| DALLE3 (Betker et al., 2023) | - | 0.96 | 0.87 | 0.47 | 0.83 | 0.43 | 0.45 | 0.67 |
| SD3-Medium (Rombach et al., 2022b) | Diff. | 0.99 | 0.94 | 0.72 | 0.89 | 0.33 | 0.60 | 0.74 |
| SANA-1.5 (Xie et al., 2025) | Diff. | 0.99 | 0.93 | 0.86 | 0.84 | 0.59 | 0.65 | 0.81 |
| **Und. and Gen.** | | | | | | | | |
| Chameleon (Team, 2024) | AR | - | - | - | - | - | - | 0.39 |
| LWM (Liu et al., 2024a) | AR | 0.93 | 0.41 | 0.46 | 0.79 | 0.09 | 0.15 | 0.47 |
| SEED-X (Ge et al., 2024) | AR | 0.97 | 0.58 | 0.26 | 0.80 | 0.19 | 0.14 | 0.49 |
| Show-o (Xie et al., 2024) | AR+Discrete Diff. | 0.95 | 0.52 | 0.49 | 0.82 | 0.11 | 0.28 | 0.53 |
| Transfusion (Zhou et al., 2024) | AR+Diff. | - | - | - | - | - | - | 0.63 |
| D-DiT (Li et al., 2024f) | Discrete Diff.+Diff. | 0.97 | 0.80 | 0.54 | 0.76 | 0.32 | 0.50 | 0.65 |
| ILLUME (Wang et al., 2024a) | AR | 0.99 | 0.86 | 0.45 | 0.71 | 0.39 | 0.28 | 0.61 |
| Janus (Wu et al., 2024a) | AR | 0.97 | 0.68 | 0.30 | 0.84 | 0.46 | 0.42 | 0.61 |
| Harmon (Wu et al., 2025) | AR | 0.99 | 0.86 | 0.66 | 0.85 | 0.74 | 0.48 | 0.76 |
| Janus-Pro (Chen et al., 2025c) | AR | 0.99 | 0.89 | 0.59 | 0.90 | 0.79 | 0.66 | 0.80 |
| Tar (Han et al., 2025) | AR | 0.99 | 0.91 | 0.76 | 0.81 | 0.57 | 0.51 | 0.76 |
| MMaDA (Yang et al., 2025) | Discrete Diff. | 0.99 | 0.76 | 0.61 | 0.84 | 0.20 | 0.37 | 0.63 |
| FUDOKI (Wang et al., 2025) | DFM | 0.96 | 0.85 | 0.56 | 0.88 | 0.68 | 0.67 | 0.77 |
| **NExT-OMNI** | DFM | 0.99 | 0.92 | 0.79 | 0.85 | 0.78 | 0.74 | 0.85 |

**Video Understanding.** We evaluate the video understanding capabilities of NExT-OMNI on several benchmarks, including MSVD-QA (Chen & Dolan, 2011), MSRVTT-QA (Xu et al., 2017),TGIF-QA (Li et al., 2016), and ActivityNet-QA (Caba Heilbron et al., 2015). As shown in Table 14, compared to models in understanding only or unified understanding-generation, NExT-OMNI achieves superior performance across all metrics, demonstrating that the DFM-based framework possesses considerable capability in understanding spatiotemporal relationships.

**Video Generation.** For video generation, we evaluate NExT-OMNI on VBench (Huang et al., 2024) and compare it against classical approaches, including Open-Sora (OpenAI, 2025), VILA-U (Wu et al., 2024c), and CogVideo (Hong et al., 2022). The results presented in Table 15 demonstrate that our method achieves superior performance compared to these autoregressive (AR)-based classical methods, highlighting the potential of discrete flow matching (DFM) in short video generation.

Table 12: **Visual Generation Results on DPG Bench (Hu et al., 2024).** "Und." and "Gen." denote the abbreviations of "Understanding" and "Generation".

| Model | Paradigm | Global | Entity | Attribute | Relation | Other | Overall↑ |
|---|---|---|---|---|---|---|---|
| **Gen. Only** | | | | | | | |
| SDv1.5 (Rombach et al., 2022a) | Diff. | 74.63 | 74.23 | 75.39 | 73.49 | 67.81 | 63.18 |
| PixArt-$\alpha$ (Chen et al., 2024a) | Diff. | 74.97 | 79.32 | 78.60 | 82.57 | 76.96 | 71.11 |
| Emu3-Gen (Wang et al., 2024b) | AR | 85.21 | 86.68 | 86.84 | 90.22 | 83.15 | 80.60 |
| SDXL (Podell et al., 2023) | Diff. | 83.27 | 82.43 | 80.91 | 86.76 | 80.41 | 74.65 |
| Playground v2.5 (Li et al., 2024b) | - | 83.06 | 82.59 | 81.20 | 84.08 | 83.50 | 75.47 |
| PixArt-$\Sigma$ (Chen et al., 2024a) | AR | 86.89 | 82.89 | 88.94 | 86.59 | 87.68 | 80.54 |
| DALLE3 (Betker et al., 2023) | Diff. | 90.97 | 89.61 | 88.39 | 90.58 | 89.83 | 83.50 |
| SD3-Medium (Rombach et al., 2022b) | Diff. | 87.90 | 91.01 | 88.83 | 80.70 | 88.68 | 84.08 |
| **Und. and Gen.** | | | | | | | |
| Show-o (Xie et al., 2024) | AR+Discrete Diff. | - | - | - | - | - | 67.48 |
| TokenFlow-XL (Qu et al., 2024) | AR | 78.72 | 79.22 | 81.29 | 85.22 | 71.20 | 73.38 |
| Janus (Wu et al., 2024a) | AR | 82.33 | 87.38 | 87.70 | 85.46 | 86.41 | 79.68 |
| Janus-Pro (Chen et al., 2025c) | AR | 87.58 | 88.63 | 88.17 | 88.98 | 88.30 | 82.63 |
| BLIP-3o (Chen et al., 2025a) | AR+Diff. | - | - | - | - | - | 81.60 |
| Tar (Han et al., 2025) | AR | 83.59 | 89.35 | 86.91 | 93.50 | 80.80 | 82.96 |
| MMaDA (Han et al., 2025) | Discrete Diff. | 77.81 | 78.48 | 81.74 | 84.79 | 63.20 | 69.97 |
| FUDOKI (Wang et al., 2025) | DFM | 80.55 | 89.73 | 88.05 | 93.66 | 78.00 | 83.63 |
| **NExT-OMNI** | DFM | 81.09 | 89.76 | 88.36 | 94.37 | 81.63 | 84.46 |

Table 13: **Comparison with state-of-the-art methods on speech-language and audio-language benchmarks.** Here, "T", "S", and "A" represent text, speech (belonging to the audio modality) audio inputs, respectively.

| Model | Librispeech (EN-WER) | | | | AudioCaps | |
|---|---|---|---|---|---|---|
| | Test_clean | | Test_other | | Test | |
| | S→T | T→S | S→T | T→S | A→T | T→A |
| **Speech Only** | | | | | | |
| SpeechT5 (Ao et al., 2021) | 2.4 | - | 5.8 | - | - | - |
| SALMONN (Sun et al., 2024a) | 2.1 | - | 4.9 | - | - | - |
| Mini-Omni (Xie & Wu, 2024) | 4.7 | - | 9.4 | - | - | - |
| Freeze-Omni (Wang et al., 2024c) | 3.2 | - | 7.7 | - | - | - |
| Qwen2-Audio (Chu et al., 2023) | 2.0 | - | 4.5 | - | - | - |
| **Omnimodal Und.** | | | | | | |
| VITA (Fu et al., 2024) | 8.1 | - | 18.4 | - | - | - |
| EMOVA (Chen et al., 2024b) | 4.0 | 3.4 | - | - | - | - |
| VITA 1.5 (Fu et al., 2024) | 3.4 | - | 7.5 | - | - | - |
| OpenOmni (Luo et al., 2025a) | 3.1 | 3.4 | 7.0 | 7.8 | - | - |
| Stream-Omni (Zhang et al., 2025b) | 3.0 | - | 7.2 | - | - | - |
| **Omnimodal Und. and Gen.** | | | | | | |
| AnyGPT (Zhan et al., 2024) | 8.5 | - | - | - | - | - |
| UnifiedIO2-xxlarge (Lu et al., 2024) | - | - | - | - | 48.9 | 2.64 |
| OmniFlow (Li et al., 2025c) | - | - | - | - | 78.4 | 1.75 |
| NExT-GPT (Wu et al., 2023) | - | - | - | - | 81.3 | 1.74 |
| **NExT-OMNI** | 3.0 | 3.1 | 7.0 | 7.2 | 84.8 | 1.65 |

Table 14: **Comparison with state-of-the-art methods on the video understanding benchmarks.**

| Model | MSVD-QA | MSRVTT-QA | TGIF-QA | ActivityNet-QA |
|---|---|---|---|---|
| **Und. Only** | | | | |
| Video-Chat (Li et al., 2023a) | 56.3 | 45.4 | 34.4 | - |
| VideoLLaMA (Zhang et al., 2023b) | 51.6 | 29.6 | - | - |
| Video-ChatGPT (Maaz et al., 2023) | 64.9 | 49.3 | 51.4 | 35.2 |
| Video-LLava (Lin et al., 2023) | 70.7 | 59.2 | 70.0 | 45.3 |
| **Und. and Gen.** | | | | |
| UnifiedIO2 (Lu et al., 2024) | 52.1 | 42.5 | | |
| NExT-GPT (Wu et al., 2023) | 64.5 | 61.4 | - | - |
| Emu (Sun et al., 2023) | - | 18.8 | 8.3 | - |
| Emu2 (Sun et al., 2024b) | 31.4 | 28.7 | - | - |
| SEED-LLaMA (Ge et al., 2023b) | 40.9 | 30.8 | - | - |
| VILA-U (Wu et al., 2024c) | 73.4 | 58.9 | 51.3 | 51.6 |
| **NExT-OMNI** | 76.2 | 62.7 | 58.1 | 56.4 |

Table 15: **Comparison with state-of-the-art methods on video generation benchmarks VBench (Huang et al., 2024).**

| Model | Total Score | Quality Score | Semantic Score |
|---|---|---|---|
| Open-Sora (OpenAI, 2025) | 75.9 | 78.8 | 62.3 |
| CogVideo (Hong et al., 2022) | 67.0 | 72.1 | 46.8 |
| VILA-U (Wu et al., 2024c) | 74.0 | 76.3 | 65.0 |
| **NExT-OMNI** | 80.1 | 80.8 | 77.5 |

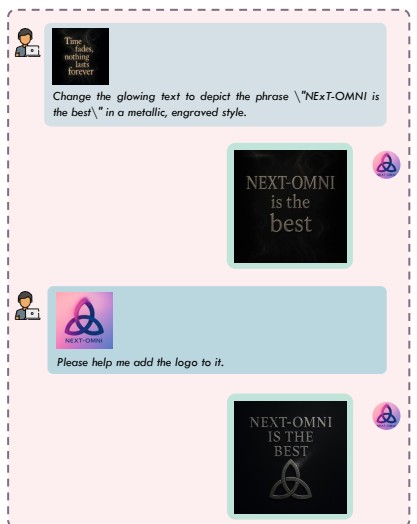 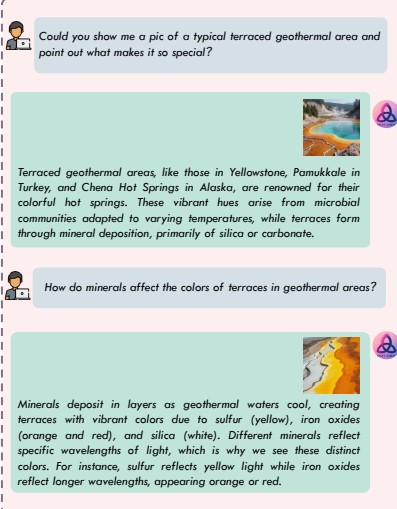

Figure 8: **Visualization case of multi-turn vision-language interaction.** NExT-OMNI is capable of performing multi-turn image generation based on predefined image generation locations (left) as well as spontaneously determining image generation locations to achieve natural multimodal interactions in a multi-turn dialogue setting (right).

# I ADDITIONAL VISUALIZATION

In addition to the above experimental results, we provide additional visualization cases to supplement the demonstration of NExT-OMNI's extensive application scenarios and interesting properties.

**Interesting Properties.** We demonstrate NExT-OMNI's iterative refinement inference process across different modal data in Figure 11. Figure 16 presents image generation quality comparisons with other similar models. Figure 21 showcases zero-shot cross-modal generalizability, capable of accepting arbitrary data inputs and generating relevant outputs in other modalities. Figure 13 illustrates discrete flow matching's single forward pass extraction of unified representations for cross-modal retrieval. Moreover, as shown in Figure 22, NExT-OMNI can spontaneously perform "think with images" without relying on external tools. By unlocking image generation during the reasoning process, it can better solve complex tasks and improve performance, demonstrating the considerable potential of NExT-OMNI.

**High-Level Multi-Turn Interaction.** We demonstrate multi-turn visual interaction capabilities in Figure 8, where the model can autonomously perceive and select appropriate positions for image generation, or manually control image generation positions. Figure 9 showcases multi-turn speech interaction capabilities, where the model can accept speech inputs and produce speech outputs.

**Basic Single-Turn Interaction.** We also supplement visualization cases of NExT-OMNI's single-turn interactions, including image generation in Figure 14, audio generation in Figure 15, video generation in Figure 17, and omnimodal understanding in Figure 10, image understanding in Figure 18, audio understanding in Figure 19, and video understanding in Figure 20. Overall, NExT-OMNI can accomplish fundamental single-turn multimodal interactions with strong capabilities.

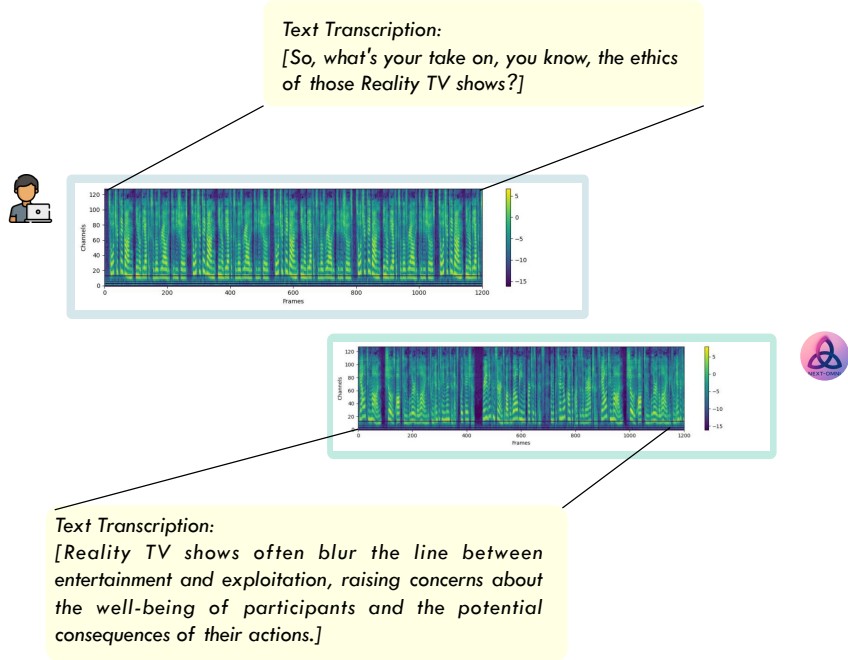

Figure 9: **Visualization case of multi-turn speech-language interaction.** NExT-OMNI is capable of flexibly using text or speech as input and output to facilitate multi-turn interactions.

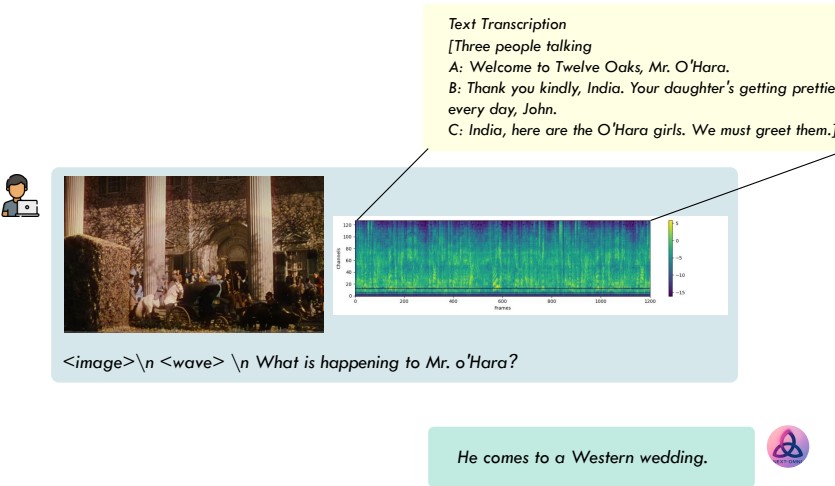

Figure 10: **Visualization case of omnimodal understanding.** NExT-OMNI demonstrates exceptional general capabilities in omnimodal understanding by effectively comprehending and responding to complex questions that encompass visual, textual, and audio information.

## J    THE USE OF LARGE LANGUAGE MODELS

We declare that large language models (LLMs) were employed to assist with the refinement of this manuscript, specifically, for grammar checking, language polishing, and improving the clarity and fluency of the text. Additionally, LLMs were used in a limited capacity for minor debugging and syntactic correction of code snippets included in the work.

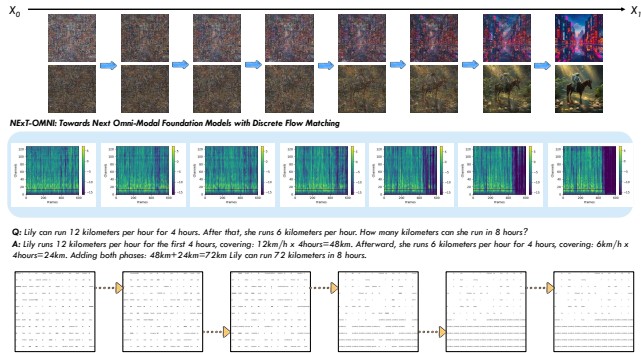

Figure 11: **Visualization case of iterative refinement process.** NExT-OMNI enables the generation of data in any modality by iteratively performing denoising through bidirectional encoding and refinement of random inputs.

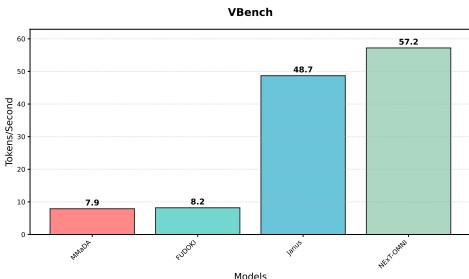

Figure 12: **Visualization case of speed comparison on VBench (Huang et al., 2024).** We select several classic methods and tested the generation speeds of these models under cached settings on VBench for 8-frame video generation. It can be observed that NExT-OMNI achieves significantly better acceleration compared to the AR-based Janus, verifying the advantage of DFM's parallel decoding mechanism in inference speed.

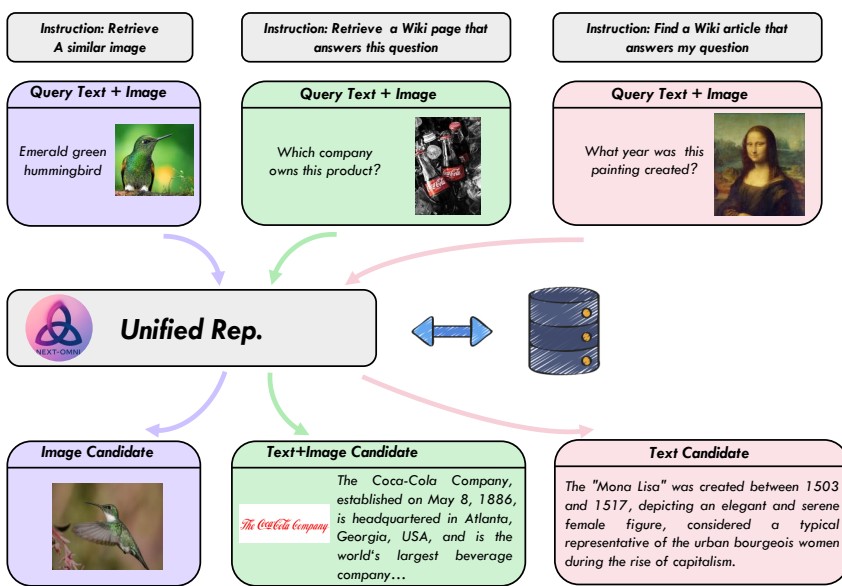

Figure 13: **Visualization case of multimodal retrieval based on unified representation.** NExT-OMNI extracts deeply fused unified representations of multimodal data through a single forward, enabling cross-modal retrieval. This more generalized capability highlights the superiority of the DFM structure compared to AR in handling multimodal tasks.

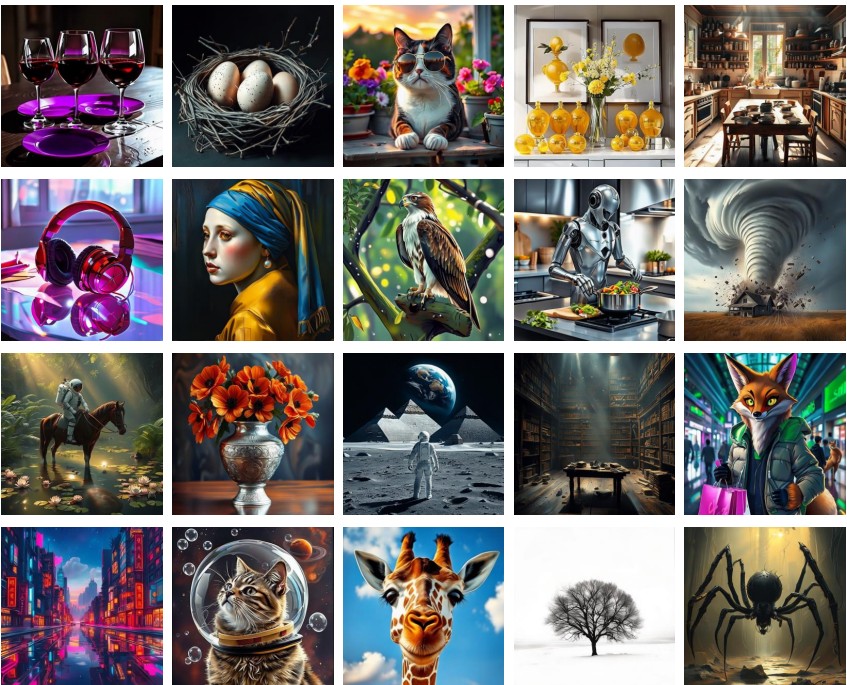

Figure 14: **Visualization case of text to image generation sampled from GenEval (Ghosh et al., 2023).** NExT-OMNI leverages a discrete flow matching mechanism to accomplish text-to-image generation tasks that adhere to both aesthetic and semantic alignment.

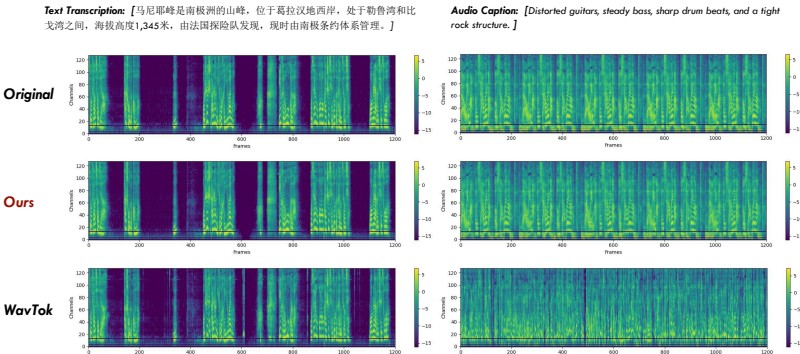

Figure 15: **Visualization case of audio generation.** NExT-OMNI utilizes a discrete flow matching mechanism to accomplish various types of audio generation tasks, including speech synthesis and music generation.

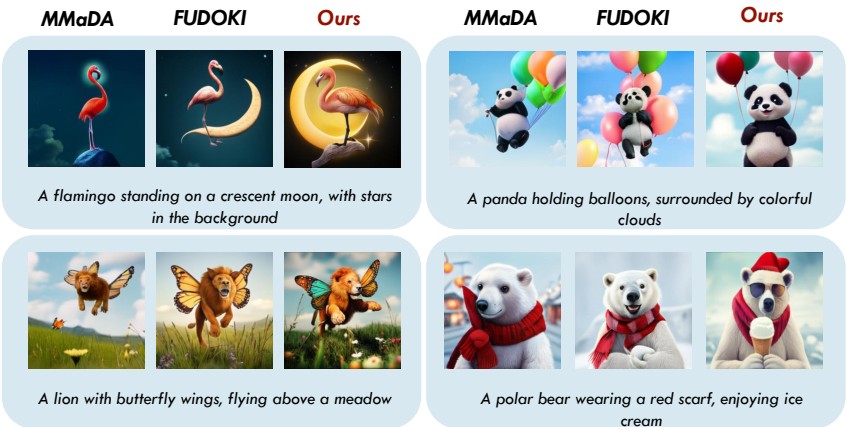

Figure 16: **Visualization case of image quality comparison.** Compared with FUDOKI (Wang et al., 2025) and MMaDA Yang et al. (2025) on various text prompts, NExT-OMNI achieves superior text-image alignment and aesthetics.

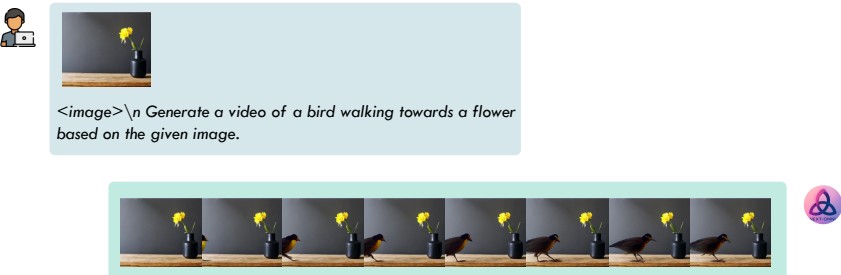

Figure 17: **Visualization case of video generation.** NExT-OMNI employs a discrete flow matching mechanism to generate short videos, demonstrating outstanding scalability.

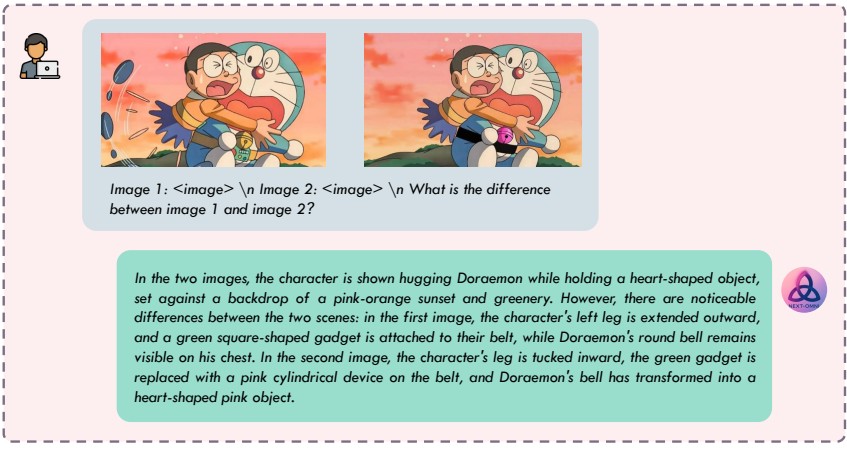

Figure 18: **Visualization case of image understanding.** As we can see from the additional qualitative examples, NExT-OMNI consistently delivers more accurate and well-aligned reasoning response on visual question answering tasks.

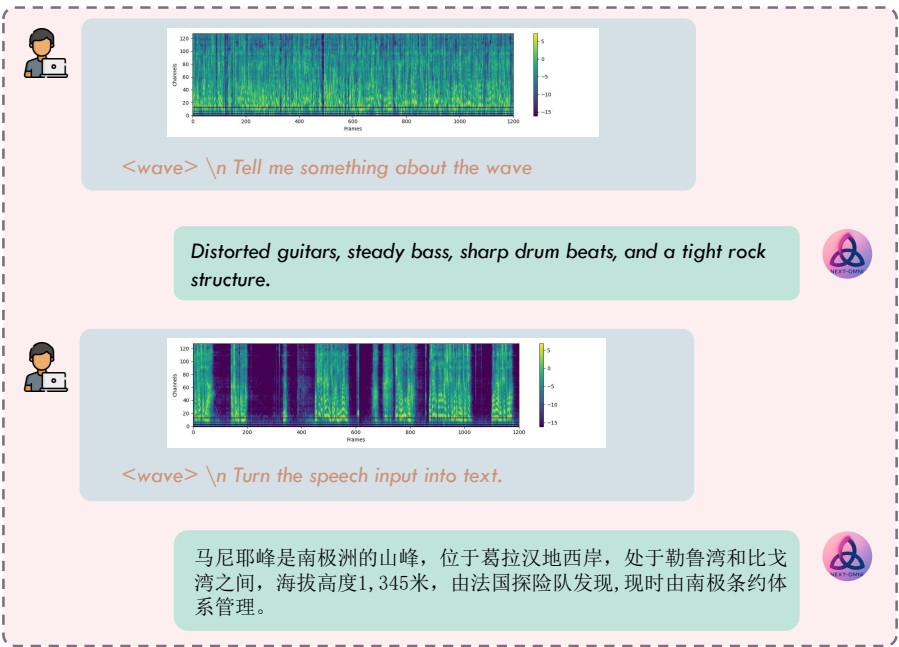

Figure 19: **Visualization case of audio understanding.** NExT-OMNI is capable of performing audio to text tasks, including speech transcription and audio to caption generation, showcasing strong generalization abilities.

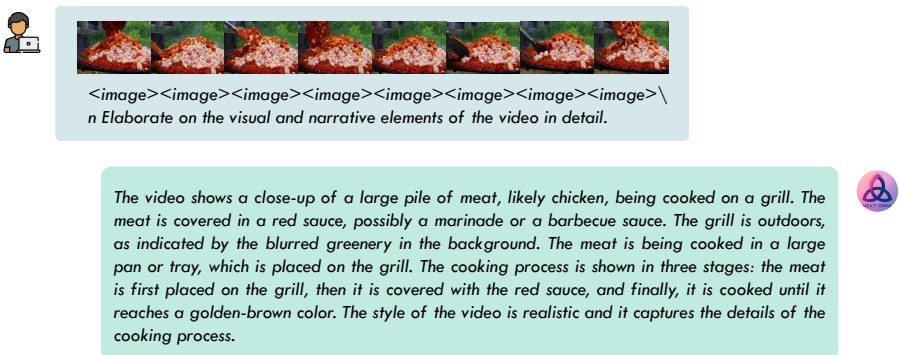

Figure 20: **Visualization case of video understanding.** NExT-OMNI is capable of processing video inputs, demonstrating spatiotemporal perception abilities.

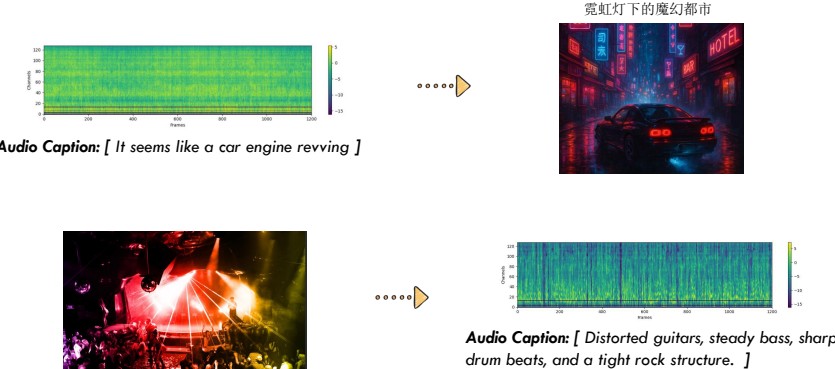

Figure 21: **Visualization case of cross-modal generation.** NExT-OMNI exhibits strong zero-shot cross-modal generation capabilities, enabling the generation of outputs in various modalities based on inputs from different modalities. This any-to-any generation ability highlights its generalized advantage in achieving deep alignment across all modalities.

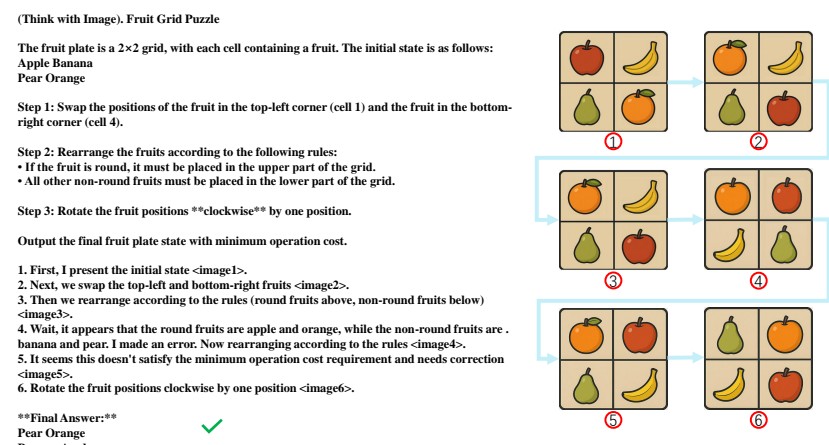

Figure 22: **Visualization case of thinking with images.** NExT-OMNI can enhance its reasoning ability through the 'thinking with images' mode, demonstrating strong potential in integrating visual cues into intermediate reasoning steps, which improves interpretability and enables more accurate problem-solving across complex multimodal tasks.

