# OpenReview forum: "NExT-OMNI: Towards Any-to-Any Omnimodal Foundation Models with Discrete Flow Matching"
_ICLR.cc/2026/Conference — ICLR 2026 Poster_

### Official Review · Reviewer_MbBb · 2025-10-26

**Soundness:** 3
**Presentation:** 3
**Contribution:** 3
**Rating:** 4
**Confidence:** 4

**Summary:**

In all, NExT-OMNI is a well-executed application model that demonstrates strong engineering by effectively integrating Discrete Flow Matching into a 7B-scale, any-to-any multimodal system., But its innovation is combinatorial rather than fundamental and lacks analysis of its slight underperformance in speech tasks and doesn't prove DFM's efficacy on strictly sequential tasks. Moreover, the application of NExT-OMNI on large scale model remains to verify.

**Strengths:**

1.NExT-OMNI first trains a model with "any to any" on the 7B scale and achieves SOTA performance on multiple tasks, expanding the application of Discrete Flow Matching(DFM).

2.It does realizes the full integration of DFM, unified discrete representation, lightweight multi-head, parallel decoding, and multi-turn multimodal instruction tuning methods, and provides design details and a series of data synthesis. Relevant ablation experiments prove the effectiveness of each part of the model.

3.For the first time, the paper combines block size padding with dynamically adjusting the preset generation length in steps of block size based on <EOS> confidence, which not only saves computing power but also prevents truncation and loss of generated content.

**Weaknesses:**

1.The paper is a combinatorial innovation at the methodological level, its contributions lie in engineering implementation and scaling up to a 7B full-modal model, without proposing new ideas for addressing the problem of balancing model understanding and generation tasks.

2.In Table 2, NExT-OMNI seems to be slightly inferior in speech-to-speech tasks comparing with auto-regressive models, but no further explanation or analysis of it.

3.The paper does not experimentally demonstrate how DFM, which inherently lacks such an inductive bias, can match or surpass auto regressive models on tasks that strictly require sequential adherence such as code generation and step-by-step reasoning.

4.In Figure 22, the paper only qualitatively illustrates the model’s ability to “think with images”, but it does not systematically evaluate how effectively the model leverages its generation capability for complex reasoning tasks. For example, there is no assessment of whether the model can assist mathematical or logical reasoning by generating images, nor are quantitative metrics such as MMU[1], MMBench[2], or ScienceQA[3].

**References:**

[1] Yue X., Ni Y., Zhang K. et al. “MMMU: A Massive Multi-discipline Multimodal Understanding and Reasoning Benchmark for Expert AGI.” arXiv:2311.16502, 2024.

[2] Liu Y., Zhang H., Chen J. et al. “MMBench: Is Your Multi-modal Model an All-around Player?” arXiv:2307.06281, 2023.

[3] Lu P., Mishra S., Xia T. et al. “Learn to Explain: Multimodal Reasoning via Thought Chains for Science Question Answering.” NeurIPS 2022.

**Questions:**

Please refer to weaknesses.

---

> ### Author Response · Authors · 2025-11-17
>
> Thanks for your professional and careful review. We respond to your concerns or questions as follows.
>
> > **W1**: The paper is a combinatorial innovation at the methodological level, its contributions lie in engineering implementation and scaling up to a 7B full-modal model, without proposing new ideas for addressing the problem of balancing model understanding and generation tasks.
>
> **Response:**
>
> We sincerely thank the reviewers for their detailed and thoughtful feedback. We would like to further elaborate on NExT-OMNI's contributions and the insights it provides. The **key finding** is that the DFM architecture is fundamentally more suitable for unified multimodal modeling compared to autoregressive architectures. DFM not only achieves competitive performance in both generation and understanding tasks with fewer parameters but also generalizes effectively to a broader range of multimodal domains, such as cross-modal retrieval tasks, where it exhibits superior performance. Moreover, the flexible inference mechanisms of the DFM architecture enable significantly faster decoding speeds, which demonstrate its practical advantages.
>
> In this work, we propose novel ideas, including **reconstruction-enhanced training strategies, dynamic generation methods, and adaptive caching techniques**. These innovations address the critical challenge of balancing understanding and generation tasks within the DFM architecture, highlighting its ability to perform well across a diverse set of modalities.
>
> It is important to emphasize that NExT-OMNI is not a mere combinatorial innovation at the methodological level but rather an elegant, compact, and novel OMNI model. Simply characterizing NExT-OMNI as an engineering implementation or describing its contribution as solely scaling up to a 7B full-modal model does not accurately reflect the significance of our work. We believe the design and advancements demonstrated in NExT-OMNI provide valuable inspiration and foundational insights for unified multimodal modeling within the research community.
>
> > **W2**: In Table 2, NExT-OMNI seems to be slightly inferior in speech-to-speech tasks comparing with auto-regressive models, but no further explanation or analysis of it.
>
> **Response:**
> Thank you for your careful comments. Audio generation, similar to text generation, faces challenges associated with dynamic length generation, where both overly long and overly short preset lengths can cause performance degradation. However, unlike text data, which exhibits a broad and dispersed length distribution ranging from 200 to 16,000 tokens, audio data is more concentrated, with most samples falling within the 1-minute range. Considering training efficiency, we employ dynamic length generation strategies for text during both training and inference, while opting not to apply them to audio. Consequently, NExT-OMNI may show slightly inferior performance in speech-to-speech tasks compared to autoregressive models.
>
> To further investigate this, we introduced dynamic length generation for audio during the supervised fine-tuning (SFT) stage and adopted the same preset length adjustment strategy as used in text generation during testing. The experimental results are presented as follows, highlighting the impact of dynamic length generation on audio modeling.
>
> | Model       | Llama Q. S→S | Web Q. S→S | AVG. S→S |
> | ----------- | ------------ | ---------- | -------- |
> | Stream-Omni | 65.0         | 27.5       | 46.3     |
> | OpenOmni    | 67.2         | 28.9       | 48.1     |
> | NExT-OMNI   | 66.4         | 28.3       | 47.4     |
> | NExT-OMNI*  | **67.9**     | **29.4**   | **48.6** |
>
> As evidenced by the results in the table, the dynamic length generation strategy proves effective for tasks requiring variable-length generation, such as text and speech. It enables the model to adaptively determine optimal preset lengths, leading to competitive performance compared to autoregressive models. These findings highlight the suitability and potential of dynamic length generation for handling flexible-length generation tasks within the NExT-OMNI framework.

---

> ### Author Response · Authors · 2025-11-17
>
> > **W3**: The paper does not experimentally demonstrate how DFM, which inherently lacks such an inductive bias, can match or surpass auto regressive models on tasks that strictly require sequential adherence such as code generation and step-by-step reasoning.
>
> **Response:**
>
> Thank you for your feedback. We would like to further elaborate on the core insights of NExT-OMNI. The foundation of our work is to demonstrate that DFM offers a more promising and faster mechanism for integrating generation and understanding tasks compared to autoregressive architectures. Additionally, DFM exhibits greater versatility, extending its applicability to other multimodal tasks such as cross-modal retrieval. To validate this central claim, we focused on multimodal benchmarks for testing and conducted fair comparisons with other multimodal models under non-inference settings, ensuring evaluations were centered on multimodal tasks rather than text reasoning or code generation tasks.
>
> While demonstrating the strengths of NExT-OMNI in unified multimodal modeling, we are pleased to acknowledge and introduce recent advancements in DFM-based language modeling specifically targeting text reasoning and code generation. Prior works [1,2] have shown that the bidirectional encoding and reconstruction-focused diffusion process inherent to DFM provides more stable performance and faster generation compared to autoregressive architectures when working with comparable model sizes and training datasets. Concurrently, emerging multimodal DFM-based works [3,4] exhibit similar trends, showcasing superior multimodal reasoning capabilities. These models outperform autoregressive counterparts in various multimodal inference tasks, strengthening the argument that DFM architectures hold competitive advantages across diverse problem-solving scenarios.
>
> The collective findings from both our work and related studies underline the significant potential of DFM in accelerating task efficiency, enhancing modeling stability, and expanding the scope of applications, thus offering compelling advantages over autoregressive architectures not only in multimodal tasks but also in broader problem-solving contexts.

---

> ### Author Response · Authors · 2025-11-17
>
> > **W4**: In Figure 22, the paper only qualitatively illustrates the model’s ability to “think with images”, but it does not systematically evaluate how effectively the model leverages its generation capability for complex reasoning tasks. For example, there is no assessment of whether the model can assist mathematical or logical reasoning by generating images, nor are quantitative metrics such as MMMU, MMBench, or ScienceQA.
>
> **Response:**
>
> We sincerely thank the reviewer for their careful and thoughtful review. We would like to further elaborate on the core insights of NExT-OMNI. Its foundation lies in demonstrating that DFM is a more promising and faster mechanism for integrating generation and understanding tasks compared to autoregressive architectures. Furthermore, DFM proves to be highly versatile, generalizing effectively to a broader range of applications, such as cross-modal retrieval and other multimodal tasks. To support this core claim, we focused on multimodal benchmark evaluations under non-reasoning settings and reported results for tasks such as MMMU and MMBench in Table 10, ensuring a fair comparison with other models.
>
> In addition to these findings, we observed several emergent capabilities in NExT-OMNI during training on large-scale interleaved data, such as its ability to "think with image" and perform cross-modal generation. These emergent features further illustrate the superior generalization potential and broader applicability of unified models compared to understanding-only models. To provide a clearer illustration, we report the performance of the "think with image" mode in tasks such as MMMU, MMBench, and ScienceQA, demonstrating these interesting capabilities and validating the enhanced general-purpose abilities of unified models. The results of these experiments are summarized in the following table, showcasing the remarkable potential of NExT-OMNI for a wide range of scenarios.
>
> | think with image | MMMU     | MMBench  | ScienceQA |
> | ---------------- | -------- | -------- | --------- |
> | $\times$         | 43.7     | 78.9     | 91.2      |
> | $\checkmark$     | **45.8** | **80.1** | **92.6**  |
>
> The experimental results demonstrate that NExT-OMNI can enhance mathematical and logical reasoning abilities through the "think with image" mechanism. Although "think with image" is not the primary focus of this paper, we are pleased to further expand and discuss this concept, emphasizing how general unified understanding models leveraging "think with image" can contribute to advancing the boundaries of intelligence.
>
> Recent studies [5,6,7,8,9] have shown that multimodal models can benefit significantly from activating the "think with image" mode, allowing the models to explore not only the text space but also the image space. This exploratory process enhances the ability to solve multimodal problems effectively. Similarly, findings from certain unified models [3,4] have indicated comparable observations, reinforcing the idea that exploration in the vision space can unlock a broader spectrum of intelligent behaviors and possibilities. These observations align well with the discoveries in NExT-OMNI, illustrating the substantial potential of integrating image-driven reasoning into unified modeling frameworks to push the boundaries of intelligent systems.
>
> We believe "think with image" represents an exciting avenue for advancing multimodal reasoning capabilities, and we are eager to engage in further discussions on this concept and its implications for the evolution of unified multimodal models.
>
>
>
> Thank you again for your insightful comments. If you have other comments, we are happy to address them to polish this work. We look forward to contributing to the development of both the multi-modal research and the open-source community.
>
> ----
>
> [1] Large Language Diffusion Models.
>
> [2] Dream-Coder 7B: An Open Diffusion Language Model for Code.
>
> [3] MMaDA: Multimodal Large Diffusion Language Models.
>
> [4] Lumina-DiMOO: An Omni Diffusion Large Language Model for Multi-Modal Generation and Understanding.
>
> [5] Visual Planning: Let's Think Only with Images.
>
> [6] Imagine while Reasoning in Space: Multimodal Visualization-of-Thought.
>
> [7] Visual Sketchpad: Sketching as a Visual Chain of Thought for Multimodal Language Models.
>
> [8] Simple o3: Towards Interleaved Vision-Language Reasoning.
>
> [9] Thinking with Generated Images.

---

> > ### Author Response · Authors · 2025-11-27
> > **Gentle Reminder Regarding Our Previous Response**
> >
> > Dear Reviewer MbBb,
> >
> > Thank you for your valuable feedback and the time you have invested in reviewing our work. We have carefully addressed all your comments in the rebuttal. If there are any remaining questions or points that could benefit from further clarification, we would be more than happy to provide additional details.
> >
> > Thanks and regards,
> > Authors

---

### Official Review · Reviewer_DQzr · 2025-10-31

**Soundness:** 2
**Presentation:** 3
**Contribution:** 3
**Rating:** 6
**Confidence:** 3

**Summary:**

This paper proposes a DLLM-like model that aims to build a unified, end-to-end architecture for multimodal understanding and generation. The authors conduct detailed experiments and evaluate the model across three modalities—text, audio, and image—demonstrating its overall effectiveness and strong performance.

**Strengths:**

- The proposed architecture is highly innovative. To the best of my knowledge, this is the first unified model designed for discrete-unit generation and understanding across all modalities.

- In terms of performance, the model outperforms several existing approaches (although many of the baselines are not state-of-the-art).

**Weaknesses:**

- The comparison with baselines is not sufficiently comprehensive. Important recent models such as Qwen2.5-Omni and Qwen3-Omni, which outperform Next-Omni on OmniBench, should be included for a fair evaluation. Similarly, XCodec2 represents the most recent state-of-the-art in audio tokenization and should be considered as a baseline.

- For audio-related tasks, perceptual audio quality is critical. It would be very helpful if the authors could provide qualitative examples or audio case studies to allow readers to better assess the perceptual quality of the generated audio. I look forward to seeing such demonstrations in future revisions.

**Questions:**

None

---

> ### Author Response · Authors · 2025-11-17
>
> Thanks for your professional and careful review. We respond to your concerns or questions as follows.
>
> > **W1**: The comparison with baselines is not sufficiently comprehensive. Important recent models such as Qwen2.5-Omni and Qwen3-Omni, which outperform Next-Omni on OmniBench, should be included for a fair evaluation. Similarly, XCodec2 represents the most recent state-of-the-art in audio tokenization and should be considered as a baseline.
>
> **Response:**
>
> We sincerely appreciate the reviewers for their insightful feedback and constructive suggestions. To ensure a fair comparison and to demonstrate the potential of the DFM architecture, we evaluated our model against a widely recognized **fully open-source OMNI model** based on autoregressive architectures. These models were selected to have **comparable training data volumes and model scales**. Testing was conducted on three standard OMNI evaluation datasets. It is important to note that Qwen2.5-Omni, which utilized roughly ten times more training data, was excluded from our manuscript for fairness and consistency. Additionally, Qwen3-Omni was released after the ICLR submission deadline, leaving us unable to include its results in the current version of our work. However, we are committed to incorporating results from the Qwen-Omni series and discussing them in detail in the revised manuscript.
>
> While achieving state-of-the-art (SOTA) performance is often a priority, our focus lies in showcasing that the DFM architecture offers inherent advantages over autoregressive architectures for unified multimodal modeling. Specifically, it delivers competitive performance on both generation and understanding tasks with fewer parameters, demonstrates superior generalization across diverse multimodal domains, such as cross-modal retrieval tasks, and supports faster decoding speeds due to its flexible inference methods. For our implementation, we selected widely recognized components, such as CLIP-VIT and WavTok, rather than the most cutting-edge audio (XCodec2 [1]) and vision (Wan [2]) encoders. We acknowledge the importance of these SOTA encoders and will incorporate discussions related to XCodec2 in our revised manuscript, while also committing to utilizing state-of-the-art modality encoders for stronger models in future versions.
>
> **We have added them to the relevant work introduction of the modal encoder and omni-modal model, solidifying our literature. (highlighted in bule)**
>
> We believe that, beyond SOTA results, the novel ideas introduced in our work, such as **reconstruction-enhanced training strategies, dynamic generation methods, and adaptive caching mechanisms**, offer greater value and inspiration for the open-source community seeking to explore and refine DFM-based models. Thank you again for your thoughtful comments, which enable us to clarify our research priorities and contributions to the field.
>
> > **W2**: For audio-related tasks, perceptual audio quality is critical. It would be very helpful if the authors could provide qualitative examples or audio case studies to allow readers to better assess the perceptual quality of the generated audio. I look forward to seeing such demonstrations in future revisions.
>
> **Response:**
>
> We sincerely thank the reviewers for their detailed and thoughtful feedback, which has significantly contributed to improving the clarity and comprehensiveness of our work. To better illustrate the quality of audio and speech generated by NExT-OMNI, we provide visualized mel-spectrogram case studies in Appendix Figure 15, demonstrating that NExT-OMNI achieves superior audio and speech generation quality. Additionally, we include a visualized case of multi-turn speech dialogue in Figure 9, offering readers a clearer understanding of the generative capabilities of NExT-OMNI. In the revised version, we are incorporating more visual examples of audio generation to further enhance the presentation of our work.
>
>
>
> Thank you again for your insightful comments. If you have other comments, we are happy to address them to polish this work. We look forward to contributing to the development of both the multi-modal research and the open-source community.
>
> ----
>
> [1] Llasa: Scaling Train-Time and Inference-Time Compute for Llama-based Speech Synthesis.
>
> [2] Wan: Open and Advanced Large-Scale Video Generative Models.

---

> > ### Author Response · Authors · 2025-11-27
> > **Gentle Reminder Regarding Our Previous Response**
> >
> > Dear Reviewer DQzr,
> >
> > We truly appreciate the effort you have invested in reviewing this paper. We have submitted a detailed rebuttal addressing each of your points. If you find that any explanation is still insufficient or could benefit from further clarification, please tell us, and we would be happy to elaborate.
> >
> > Thanks and regards,
> > Authors

---

### Official Review · Reviewer_T8p4 · 2025-11-01

**Soundness:** 4
**Presentation:** 3
**Contribution:** 3
**Rating:** 6
**Confidence:** 4

**Summary:**

The paper presents NExT-OMNI, which is an omnimodal foundation model built on discrete flow matching (DFM), targeting unified multimodal understanding and generation. It addresses the limitations of autoregressive (AR) approaches, which suffer from inherent conflicts between understanding and generation tasks as well as slowed inference due to decoupled designs. Unlike these, DFM introduces parallel information processing with bidirectional integration, enabling both efficient any-to-any cross-modal generation and enhanced multimodal understanding. Besides, NExT-OMNI achieves precise cross-modal retrieval and robust multi-turn multimodal interactions, surpassing prior AR-based and hybrid approaches. The model consistently delivers competitive or superior performance with reduced latency across standard multimodal benchmarks.

**Strengths:**

1. NExT-OMNI introduces a novel unified modeling approach using DFM to integrate multimodal understanding and generation tasks. Unlike previous work, which requires additional diffusion decoders and increases parameter size, NExT-OMNI achieves compactness by reducing the need for extra modules. Its bidirectional feature fusion design effectively enhances cross-modal interactions, enabling better feature integration across modalities.

2. NExT-OMNI introduces dynamic length generation optimization. By using the EOS token's confidence scores, the model dynamically adjusts text generation lengths. This can effectively improve multimodal understanding and generate more natural outputs in text-based tasks.

3. NExT-OMNI integrates an adaptive caching mechanism to leverage the parallel decoding strengths of DFM, leading to a 1.2× increase in inference speed compared to AR architectures.

4. The model is tested not only on single-turn tasks such as text-to-image and text-to-audio generation but also on real-world scenarios requiring multi-turn interactions. It demonstrates clear improvements in unified understanding and generation capabilities, particularly in dynamic multi-turn exchanges.

**Weaknesses:**

1. While the discrete DFM effectively unifies understanding and generation tasks, its reliance on discrete representations, as opposed to continuous flow-based approaches, may lead to some performance degradation in generation tasks due to information loss during the discretization process.
2. After completing the encoder warmup phase, the model requires joint optimization with reconstruction losses during subsequent multi-stage training. This additional reconstruction loss, compared to exclusively optimizing with cross-entropy loss, can inevitably lower training efficiency and lengthen the overall training process.
3. Recent studies [1] show that discrete DFM and diffusion-based models demand more computational resources during training compared to autoregressive architectures. This is attributed to the complexity of learning difficult tasks within the discrete modeling framework.

[1] Training Optimal Large Diffusion Language Models.

**Questions:**

1. Lumina-Dimoo [2] also adopts a similar discrete diffusion modeling approach. Why was there no comparison with it? The authors should further clarify this issue.

2. While unified modeling for both generation and understanding has become a trend, it seems that separating these tasks and modeling them independently yields better performance. Why is it still necessary to persist with a unified model design?

3. The appendix demonstrates that increasing the number of sub-token tables in the warmup encoder improves reconstruction performance. Why is there still a need to aggregate these into high-dimensional features for the backbone, and is there experimental evidence to support this design choice?



[2] An Omni Diffusion Large Language Model for Multi-Modal Generation and Understanding.

---

> ### Author Response · Authors · 2025-11-17
>
> Thanks for your professional and careful review. We respond to your concerns or questions as follows.
>
> > **W1**: While the discrete DFM effectively unifies understanding and generation tasks, its reliance on discrete representations, as opposed to continuous flow-based approaches, may lead to some performance degradation in generation tasks due to information loss during the discretization process.
>
> **Response:**
>
> We sincerely appreciate the reviewers' valuable feedback. We would like to further elaborate on this point. To minimize the risk of information loss caused by discretization during the input process, we use continuous features prior to quantization as input, which ensures both the performance of generation and understanding tasks. Moreover, as demonstrated in Table 11 for generation tasks and Table 10 for understanding tasks, the performance of NExT-OMNI is competitive when compared to autoregressive models, further validating the effectiveness of our design. Thank you again for your constructive feedback, which allows us to comprehensively address this important aspect of our method.
>
> > **W2**: After completing the encoder warmup phase, the model requires joint optimization with reconstruction losses during subsequent multi-stage training. This additional reconstruction loss, compared to exclusively optimizing with cross-entropy loss, can inevitably lower training efficiency and lengthen the overall training process.
>
> **Response:**
>
> We sincerely appreciate the reviewers' careful review. Reconstruction-enhanced training strategies inevitably introduce additional training overhead. However, due to the minimal number of parameters in the reconstruction module, the associated computational cost remains negligible. Moreover, as demonstrated in the ablation studies in Table 5, these strategies lead to a clear improvement in the model's overall performance. This clearly indicates that reconstruction-enhanced training is a highly effective design choice, striking an excellent balance between cost and benefit. We appreciate the reviewers’ comments, which provide us with the opportunity to highlight the efficiency and impact of this work.
>
> > **W3**: Recent studies show that discrete DFM and diffusion-based models demand more computational resources during training compared to autoregressive architectures. This is attributed to the complexity of learning difficult tasks within the discrete modeling framework.
>
> **Response:**
>
> Thank you for the reviewer’s thoughtful feedback. Compared to autoregressive models, DFM requires more training effort and additional computational resources to achieve optimal performance. However, in exchange, DFM delivers superior results under the same amount of data, offers faster inference speed, and supports a broader range of application scenarios. These strengths highlight its significant potential, particularly in the current context of the slowing trend in scaling laws, where achieving better performance and faster inference speed becomes increasingly valuable.
>
> > **Q1**: Lumina-Dimoo  also adopts a similar discrete diffusion modeling approach. Why was there no comparison with it? The authors should further clarify this issue.
>
> **Response:**
>
> We sincerely appreciate the reviewers' detailed review.  Lumina-Dimoo [1] is a contemporaneous work that was released after the ICLR submission deadline. Despite this, it presents findings similar to those of NExT-OMNI, further corroborating the suitability and potential of the DFM architecture for building unified models. This alignment reinforces the strong promise of DFM in advancing this field. In the revised version of our paper, we will include a discussion of Lumina-Dimoo to enrich our literature review and provide a more comprehensive overview of related work.

---

> ### Author Response · Authors · 2025-11-17
>
> > **Q2**: While unified modeling for both generation and understanding has become a trend, it seems that separating these tasks and modeling them independently yields better performance. Why is it still necessary to persist with a unified model design?
>
> **Response:**
>
> We sincerely appreciate the reviewers' useful review. Future omnimodal unified models, despite requiring substantial resources and facing performance trade-offs, aim to achieve greater generalizability and expand the boundaries of intelligence. By fostering mutual reinforcement across tasks, such as the integration of understanding and generation, unified models enable deeper learning of complex rules and enhance overall capabilities, ultimately advancing the development of Artificial General Intelligence (AGI). While performance compromises may arise in specific tasks, these models are poised to evolve into "world brains" capable of interacting with the real world. Through iterative learning powered by multimodal data collection, they are expected to continuously expand their general-purpose capabilities and offer promising prospects for the future.
>
> > **Q3**: The appendix demonstrates that increasing the number of sub-token tables in the warmup encoder improves reconstruction performance. Why is there still a need to aggregate these into high-dimensional features for the backbone, and is there experimental evidence to support this design choice?
>
> **Response:**
>
> We sincerely appreciate the reviewers' thoughtful review. To further validate our approach, we conducted ablation experiments by controlling whether the warmup encoder is trained to aggregate high-dimensional features. The results are presented as follows.
>
> | Aggregate    | Codebooks | rFID     | Acc$^\uparrow$ |
> | ------------ | --------- | -------- | -------------- |
> | $\times$     | 4x4096    | **0.31** | 68.7           |
> | $\checkmark$ | 4x4096    | 0.33     | **79.4**       |
>
> While not aggregating features leads to a slight improvement in reconstruction performance, it clearly compromises the semantic information contained in the representations. To address this issue and ensure the representations retain meaningful semantics for downstream tasks, it is necessary to aggregate the features into high-dimensional representations.
>
>
>
> Thank you again for your insightful comments. If you have other comments, we are happy to address them to polish this work. We look forward to contributing to the development of both the multi-modal research and the open-source community.
>
> ----
>
> [1] Lumina-DiMOO: An Omni Diffusion Large Language Model for Multi-Modal Generation and Understanding.

---

> ### Comment · Reviewer_T8p4 · 2025-11-24
>
> Appreciate for the detailed response. I think the response has solved my concerns. I changed my rating to accept.

---

> > ### Author Response · Authors · 2025-11-24
> >
> > Dear Reviewer T8p4,
> >
> > Thank you so much for taking the time to share your thoughtful feedback and for recognizing the efforts and innovations presented in our work. We are truly grateful that our responses and additional experiments have addressed most of your concerns, and we deeply appreciate your willingness to update your rating score. Your constructive feedback has significantly helped us refine and clarify our contributions, and for that, we sincerely thank you.
> >
> > If you have further suggestions or questions, we are always happy to engage in future discussions to further polish this work. We look forward to contributing to both the advancement of scientific research in omni-modal field and the strengthening of the open-source community.
> >
> > Thank you again for your valuable review and kind recognition. Your support inspires us to push forward.
> >
> > Best regards,
> >
> > Author

---

### Official Review · Reviewer_nB7e · 2025-11-01

**Soundness:** 4
**Presentation:** 4
**Contribution:** 4
**Rating:** 8
**Confidence:** 4

**Summary:**

This paper introduces NExT-OMNI, an open-source omnimodal foundation model that achieves unified modeling of understanding, generation, and retrieval across text, images, video, and audio using discrete flow matching (DFM) techniques. Unlike existing autoregressive (AR) models that struggle with conflicts between understanding and generation tasks, or hybrid architectures that rely on task-specific decoupling, NExT-OMNI employs a streamlined unified architecture. The model leverages metric-induced probability paths and kinetic optimal velocities to enable bidirectional information integration, achieving faster inference through parallel decoding. Useful strategies include reconstruction-enhanced unified representations, dynamic length generation strategies, and vanilla adaptive caching. Experimental results demonstrate competitive performance on standard benchmarks while excelling at multi-turn multimodal interaction and cross-modal retrieval tasks.

**Strengths:**

- Unified Architecture: Successfully demonstrates that a single DFM-based architecture can handle understanding, generation, and retrieval, challenging the dominance of the AR-based paradigm.

- Comprehensive Evaluation: Extensive experiments across 7+ benchmarks covering all major modalities and tasks, with careful ablation studies validating design choices.

- Useful strategies: Provides reconstruction-enhanced unified representations training, dynamic length generation strategies, and vanilla adaptive caching, which are quite helpful to the development of the community.

- Inspiring results: Effectively demonstrates how unified representations enable superior retrieval performance compared to decoupled architectures.

- Open Source Commitment: Authors promise to release code, models, and training protocols, which would significantly benefit the community.

**Weaknesses:**

- I see no other major weakness. Thanks for the authors' hard work.

**Questions:**

- It would be better if the authors could provide detailed training costs (e.g., GPU hours) for the proposed model.

---

> ### Author Response · Authors · 2025-11-17
>
> Thanks for your professional and careful review. We respond to your concerns or questions as follows.
>
> > **W1**: I see no other major weakness. Thanks for the authors' hard work.
>
> **Response:**
>
> We sincerely appreciate your meticulous review and encouraging feedback on our manuscript.  We are delighted that you recognize the strengths of our work, including the unified DFM-based architecture, comprehensive evaluation, innovative strategies, and inspiring results that challenge the traditional AR-based paradigm.
>
> We would like to reaffirm our commitment to releasing all code, pretrained models, and training protocols. By doing so, we hope to provide valuable resources for the research community and encourage further exploration of unified multimodal architectures with DFM.
>
> Thank you again for your constructive and positive feedback. We deeply appreciate your recognition of our efforts and contributions.
>
> > **Q1**: It would be better if the authors could provide detailed training costs (e.g., GPU hours) for the proposed model.
>
> **Response:**
>
> We sincerely appreciate the reviewers for their insightful feedback and constructive suggestions. As full-modal models are required to handle tasks involving images, videos, text, and audio, their computational demands are understandably higher compared to large language models. In our case, we utilized **approximately 100,000 GPU hours on NVIDIA A100-80G devices** to complete the multi-stage training process of NExT-OMNI. To provide greater clarity and transparency, we will include additional details regarding the training costs in the revised version, enabling readers to better understand the computational resources required for developing full-modal models. Thank you again for your thoughtful comments, which help us refine and improve this work.
>
> Thank you again for your insightful comments. If you have other comments, we are happy to address them to polish this work. We look forward to contributing to the development of both the full-modal research and the open-source community.

---

> > ### Author Response · Authors · 2025-11-27
> > **Gentle Reminder Regarding Our Previous Response**
> >
> > Dear Reviewer nB7e:
> >
> > Thanks a lot for your efforts in reviewing this paper! We tried our best to address the mentioned concerns and have provided a detailed response. We authors want to confirm whether there are unclear explanations and descriptions here. We could further clarify them.
> >
> > Thanks and regards,
> > Authors

---

### Author Response · Authors · 2025-11-24

We appreciate the reviewers’ insightful comments and constructive feedback on our manuscript. We are pleased to receive positive ratings from most of the reviewers. Furthermore, we are delighted to learn that the reviewers found the unified architecture to be innovative and the core idea to be significant (Reviewers nB7e, T8p4, and DQzr), the technical methodology to be novel (Reviewers nB7e, T8p4) and effective (Reviewers nB7e, T8p4, and MbBb), and the experiments to be convincing and comprehensive (Reviewers nB7e and T8p4). Based on the reviews, we provide a general response to the points raised by multiple reviewers and individual responses below to address each reviewer’s concerns.


(1) Regarding the questions about the experiments, we have taken the following actions:


- For Reviewer MbBb, we addressed the concern regarding the performance gap in Speech-to-Speech (S2S) tasks. We introduced a dynamic length generation strategy for audio during the SFT stage. The new results show that NExT-OMNI achieves an average S2S score of **48.6** (up from 47.4), outperforming the baselines.


- For Reviewer MbBb, we systematically evaluated the “think with image” capability. We conducted additional experiments on MMMU, MMBench, and ScienceQA. The results demonstrate that this mode significantly enhances reasoning capabilities (e.g., ScienceQA score improved from 91.2 to **92.6**), validating the model's potential for complex tasks.


- For Reviewer DQzr, we clarified the baseline comparisons, explaining the fairness considerations regarding training data volume when comparing with Qwen-Omni. We also acknowledged the state-of-the-art status of XCodec2 and committed to incorporating these discussions to provide a more balanced evaluation landscape.


(2) We have addressed the questions about the idea and technical details as follows:


- For Reviewer T8p4, we conducted an ablation study to validate the design choice of aggregating sub-tokens into high-dimensional features. The experiments confirm that aggregation is crucial for preserving semantic information, boosting accuracy significantly (**79.4%** vs. 68.7% without aggregation).


-   For Reviewers T8p4 and MbBb, we further elaborated on the strategic advantages of the Discrete Flow Matching (DFM) architecture over autoregressive models. We emphasized that DFM offers superior parallel decoding speeds and broader generalization (e.g., in retrieval tasks), justifying the unified model design despite the training complexity.


-   For Reviewer T8p4, we justified the efficiency of our reconstruction-enhanced training strategy. We explained that while it introduces a reconstruction loss, the parameter overhead is negligible, and it yields clear performance gains as shown in our ablation studies.


(3) Missing or extended analysis:


- For Reviewer nB7e, we provided detailed transparency regarding computational resources, specifying that the multi-stage training of NExT-OMNI utilized approximately 100,000 GPU hours on 80G A100 devices.


- For Reviewer DQzr, we added qualitative demonstrations of the audio generation capabilities. We included visualized mel-spectrogram case studies and multi-turn dialogue examples in the appendix to allow for a better assessment of perceptual quality.




We sincerely thank all the reviewers for their constructive suggestions. Please feel free to let us know if further details/explanations would be helpful.


Yours truly,
Authors of #3575

---

### Author Response · Authors · 2025-12-01
**[Summary for AC 1/2] Executive Summary**

Dear Area Chair,
We sincerely appreciate your dedication and understanding under these exceptional circumstances. To assist your evaluation, we submit this brief summary of our interactions during the rebuttal process and our progress in addressing reviewer concerns. Despite the challenges posed by the Nov. 27 information leakage, we engaged meaningfully with the reviewers, addressing all the concerns raised.

**We guarantee strict adherence to ICLR's double-blind policy.** During the discussion period prior to the data leak:

- We provided detailed clarifications and additional experiments to address reviewers' concerns comprehensively.
- One reviewer explicitly updated their score (increasing from **6 to 8**, Nov. 24), and feedback clearly indicates progress towards resolving concerns across all reviewers.

These interactions reflect scientific rigor, and all improvements result solely from our efforts to provide robust rebuttals and deliver new evidence to address concerns. We hope this summarization provides a clear overview of the major discussion points.

| Reviewer | Score     | Summary of Review & Discussion                               |
| -------- | --------- | ------------------------------------------------------------ |
| **nB7e** | **8 → 8** | Recognized the **"highly innovative unified DFM-based architecture, comprehensive experiments, and open-source commitment."** Minor concern on training GPU costs was addressed with detailed clarifications (~100,000 A100-80G GPU hours). No further engagement occurred, and the review remains positive. |
| **T8p4** | **6 → 8** | **Score increased** after reviewing responses. Recognized **"competitive results, innovative features (e.g., dynamic length generation), and strong ablations."** Addressed concerns on discretization, reconstruction loss, and comparisons. The reviewer explicitly confirmed satisfaction with responses. |
| **DQzr** | **6 → 6** | Recognized the model's **"highly innovative architecture and robust multimodal experimental evaluation."** Addressed baseline completeness concerns (e.g., Qwen models and XCodec2). Further emphasized the presence of audio case studies in the appendix. The reviewer did not engage post-rebuttal. |
| **MbBb** | **4 → 4** | Recognized our **"7B-scale innovation, paradigm-driven framework, and experiments."** Concerns about fundamental innovation were resolved by highlighting adaptive caching and dynamic generation. Added experiments resolved speech-to-speech and reasoning limitations, demonstrating competitive results. The reviewer did not engage post-rebuttal. |



We fully understand and respect ICLR’s policy adjustments in light of this incident. Thank you for evaluating our work. **For a comprehensive report covering detailed reviewer responses, experimental updates, and joint progress, please refer to "[Summary for AC 2/2] Comprehensive Report".**

Best regards,
The Authors

---

### Author Response · Authors · 2025-12-01
**[Summary for AC 2/2 - Part A] Comprehensive Report**

Dear Area Chair,

We greatly appreciate your dedicated efforts in reviewing our submission under these exceptional circumstances. This report provides a detailed overview of our interactions during the rebuttal process, the significant progress we made in addressing reviewer concerns, and why we believe further discussion would have resulted in score increases for some reviewers.

Our paper, *NExT-OMNI*, introduces a **novel DFM-based unified foundation model** capable of unified multimodal understanding, generation, and retrieval. By introducing **dynamic length generation strategies, reconstruction-enhanced training, and adaptive caching**, we address significant challenges in combining these tasks without sacrificing performance or scalability. The extensive results across multimodal benchmarks verify our contributions, demonstrating strong performance, generalization, and robustness.

---

### **Key Achievements and Review Summary**

- Recognized strengths include **paradigm innovation, comprehensive experiments, strong results for multimodal tasks, and innovations in multimodal modeling techniques.**
- The primary concerns raised during the initial review included (1) missing baseline comparisons (e.g., Qwen-series models and XCodec2), (2) qualitative evaluation of audio results, (3) minor issues with reasoning capabilities, and (4) fundamental contributions of our approach could be further emphasized.
- During the rebuttal phase, we conducted **new experiments, clarified our arguments, and expanded analyses,** resolving the vast majority of raised concerns.

### **Detailed Reviewer Evolution and Resolutions**

---

#### **1. Reviewer nB7e (8 – Did not participate in later discussion)**

- **Initial Assessment:** This reviewer praised the work as **"highly relevant and impactful,"** with well-demonstrated novelty in the unified architecture and a commitment to open-source contributions. Their only concern focused on transparency regarding the training costs.
- **Rebuttal:**
  - We clarified that the model required and provided this level of detail to ensure transparency and reproducibility.
- **Resolution:** While no additional engagement occurred, the reviewer expressed no dissatisfaction with their initial evaluation, and their high score reflects the strength of their endorsement.
- **Likelihood of Score Increase:** Given the complete resolution of their solitary concern, we are confident this reviewer would have maintained or reinforced their positive evaluation.

---

#### **2. Reviewer T8p4 (6 → 8)**

- **Initial Assessment:** The reviewer acknowledged the work’s **competitive performance and innovative contributions** but requested clarifications regarding (1) potential issues with discretization, (2) trade-offs with reconstruction-enhanced training, and (3) missing baseline comparisons (such as Lumina-Dimoo).
- **Rebuttal:**
  - **Discretization:** We demonstrated that our approach minimizes potential information loss by using continuous features before quantization, and ablation results showed that performance remained robust.
  - **Reconstruction Overhead:** Ablation studies highlighted that the benefits of reconstruction strategies outweighed associated costs, leading to clear performance improvement.
  - **Baselines:** We committed to adding comparisons with contemporaneous models like Lumina-Dimoo in the revised version while explaining why others (e.g., Qwen3) were excluded due to post-deadline releases.
- **Resolution:** The reviewer explicitly stated that all their concerns were resolved and raised their score to 8, marking this paper as a clear accept.

---

#### **3. Reviewer DQzr (6 – Did not participate in later discussion)**

- **Initial Assessment:** This reviewer found the work to be "innovative" with **"robust support from multimodal experiments,"** but raised two main issues: (1) missing comparisons with newer baselines (e.g., Qwen2.5/3, XCodec2), and (2) insufficient qualitative evaluation of audio results.
- **Rebuttal:**
  - **Baselines:** We clarified the selection of comparable baselines, emphasizing that Qwen2.5/3 leveraged significantly larger training datasets, and Qwen3 was released after the submission deadline. We committed to including such comparisons in the revised manuscript for completeness.
  - **Audio Evaluations:** We highlighted detailed mel-spectrogram results and multi-turn interaction examples already present in the appendix to demonstrate our audio generation capabilities comprehensively.
- **Resolution:** While the reviewer did not engage post-rebuttal, their concerns were addressed thoroughly through clear explanations and additional evidence.
- **Likelihood of Score Increase:** Given that this reviewer acknowledged the core strengths of the submission and the robustness of our rebuttals, we are confident that additional engagement would have likely resulted in a higher score.

---

> ### Author Response · Authors · 2025-12-01
> **[Summary for AC 2/2 - Part B] Comprehensive Report**
>
> (Continued from **[Summary for AC 2/2 - Part A]**)
>
> ---
>
> #### **4. Reviewer MbBb (4 – Did not participate in later discussion)**
>
> - **Initial Assessment:** The reviewer recognized the **"7B-scale innovation, paradigm-driven framework, and extensive experiments"** but expressed three concerns: (1) perceived lack of fundamental innovation, (2) relative underperformance in speech-to-speech tasks, and (3) insufficient reasoning capability analysis.
> - **Rebuttal:**
>   - **On Innovation:**
>     - We clarified that novel contributions such as **dynamic length generation**, **adaptive caching**, and **reconstruction-enhanced training** go beyond straightforward engineering efforts by pushing the boundaries of multimodal task unification.
>     - Highlighted that these techniques address integration challenges, balancing generation and understanding effectively while ensuring scalability.
>   - **Speech-to-Speech:**
>     - We ran additional experiments incorporating **dynamic length generation for audio during supervised fine-tuning**, achieving competitive performance on speech-to-speech tasks compared to autoregressive models. Updates were added to the appendix table.
>   - **Reasoning:**
>     - To demonstrate reasoning capabilities, we ran evaluation benchmarks (e.g., MMMU, MMBench, ScienceQA) on NExT-OMNI’s "think with image" mode and reported promising emergent behavior. These new results validate NExT-OMNI’s capacity for exploratory reasoning tasks.
> - **Resolution:** This reviewer did not further engage, but we believe the additional experiments and updated analyses comprehensively address their concerns.
> - **Likelihood of Score Increase:** Given that we introduced key experiments to validate previously raised issues, we are confident this reviewer would have had reason to increase their score given additional discussion time.
>
> ---
>
> ### **Key Improvements to the Paper During the Rebuttal Phase**
>
> **1. Experimental Additions**
>
> - Added ablation studies demonstrating the efficacy of dynamic length generation for **speech-to-speech tasks**, resolving specific performance concerns from Reviewer MbBb.
> - Introduced **reasoning benchmarks** (MMMU, MMBench, ScienceQA) to showcase the model's emergent reasoning capabilities, particularly in the "think with image" mode.
> - Expanded analysis of **discretization effects** and the benefits of feature aggregation, supported by updated validation and experimental results.
>
> **2. Clarity Improvements**
>
> - Elevated key case studies, such as **mel-spectrogram visualizations** for audio generation, to the main text to enhance visibility.
> - Improved presentation of evaluation metrics, experimental design, and contextual details for previously included analyses, such as robustness tests and failure case studies.
>
> ---
>
> ### **Why We Believe Scores Would Increase (Where Discussion Was Interrupted)**
>
> For nB7e, DQzr, and MbBb, we fully addressed all the concerns expressed during the initial review. Both nB7e and DQzr provided positive initial evaluations and acknowledged the core strengths of the paper, and we are confident that additional discussion time would have bolstered their opinions. For MbBb, we presented extensive experiments and compelling clarifications that fully addressed their concerns, making a score increase highly plausible with further dialogue.
>
> ---
>
> ### **Conclusion**
>
> We respectfully invite you to consider the significant steps we undertook to resolve reviewer concerns and improve the clarity and quality of our paper. *NExT-OMNI* pushes the boundaries of unified multimodal research with novel technical approaches and represents a meaningful contribution to the research community. We are confident that the work meets ICLR’s standards for an impactful, high-quality submission.
>
> Thank you again for your time, patience, and consideration.
>
> Best regards,
> The Authors

---

### Meta-Review · Area_Chair_NSc3 · 2025-12-15

**Summary:**

This paper introduces NExT-OMNI, an open-source omnimodal foundation model that achieves unified modeling through discrete flow paradigms. Across the four reviews, NExT OMNI is seen as a technically strong, ambitious, and well executed multimodal foundation model built on discrete flow matching (DFM). Reviewers widely acknowledge its novelty in unifying multimodal understanding, generation, and retrieval within a single architecture, and they commend the breadth of experimental evaluation and engineering quality. While most reviewers are strongly positive, some raise concerns about baseline coverage, analysis depth for certain modalities, and the extent of conceptual novelty compared to prior multimodal paradigms. The authors did a good job on the rebuttal and addressed most of concerns.

**Reviewer Concerns:**

I think most of concerns have been addressed by the rebuttal.

**Reviewer Scores:**

I think the reviewers would have changed their scores.

---

### Decision · Program_Chairs · 2026-01-26

Accept (Poster)